# Design, and dynamic evaluation of a novel photovoltaic pumping system emulation with DS1104 hardware setup: Towards innovative in green energy systems

Amir Fatah[1]*, Tarek Boutabba[1,2], Idriss Benlaloui[1]*, Said Drid[1,3], Mohamed Metwally Mahmoud[4], Mahmoud M. Hussein[4,5], Wulfran Fendzi Mbasso[6]*, Hany S. Hussein[7,8], Ahmed M. Ewias[4]

1 LSPIE Laboratory, Mostefa Ben Boulaid, University of Batna 2, Boukhlouf, Algeria, 2 University of Khenchela, El-Hamma, Khenchela, Algeria, 3 Higher National School of Renewable Energy, Environment & Sustainable Development, Batna, Algeria, 4 Electrical Engineering Department, Faculty of Energy Engineering, Aswan University, Aswan, Egypt, 5 Department of Communications Technology Engineering, Technical College, Imam Ja'afar Al-Sadiq University, Baghdad, Iraq, 6 Technology and Applied Sciences Laboratory, U.I.T. of Douala, University of Douala, Douala, Cameroon, 7 Electrical Engineering Department, College of Engineering, King Khalid University, Abha, Saudi Arabia, 8 Electrical Engineering Department, Aswan Faculty of Engineering, Aswan University, Aswan, Egypt

* fendzi.wulfran@yahoo.fr (WFM); a.fateh@univ-batna2.dz (AF); i.benlaloui@univ-batna2.dz (IB)

**Data Availability Statement:** All relevant data are within the manuscript.

**Funding:** The authors thank the Dean of Scientific Research at King Khalid University for funding this

## Abstract

Diesel engines (DEs) commonly power pumps used in agricultural and grassland irrigation. However, relying on unpredictable and costly fuel sources for DEs pose's challenges related to availability, reliability, maintenance, and lifespan. Addressing these environmental concerns, this study introduces an emulation approach for photovoltaic (PV) water pumping (WP) systems. Emulation offers a promising alternative due to financial constraints, spatial limitations, and climate dependency in full-scale systems. The proposed setup includes three key elements: a PV system emulator employing back converter control to replicate PV panel characteristics, a boost converter with an MPPT algorithm for efficient power tracking across diverse conditions, and a motor pump (MP) emulator integrating an induction motor connected to a DC generator to simulate water pump behaviors. Precise induction motor control is achieved through a controlled inverter. This work innovatively combines PV and WP emulation while optimizing system dynamics, aiming to develop a comprehensive emulator and evaluate an enhanced control algorithm. An optimized scalar control strategy regulates the water MP, demonstrated through MATLAB/Simulink simulations that highlight superior performance and responsiveness to solar irradiation variations compared to conventional MPPT techniques. Experimental validation using the dSPACE control desk DS1104 confirms the emulator's ability to faithfully reproduce genuine solar panel characteristics.

work through a general research project under grant number (RGP.2/125/45). This fund plays an important role in study design, data collection and analysis., Dr. Wulfran FENDZI MBASSO.

**Competing interests:** The authors have declared that no competing interests exist.

**Abbreviations:** $\emptyset$, Inductor flux (Wb); $C_{dc}$, Boost link capacitor ($\mu$F); $I_a$, Rotor current (A); $I_D$, Diode current (A); $I_{ph}$, $I_0$, Photo-current and the saturation current (A); $I_{sh}$, Shunt resistor current (A); $k$, Boltzmann's constant ($1.38 \times 10^{-19}$ J/K); $L_{boost}$, Boost converter inductance (mH); $n$, Ideality constant of the diode; $N_p$, Number of PV modules in parallel; $N_s$, Number of PV modules in series; $q$, Electron charge ($1.6 \times 10^{-23}$ C); $R_p$, Equivalent parallel resistance of the PV module ($\Omega$); $R_s$, Equivalent series resistance of the PV module ($\Omega$); T, Module operating temperature (K); $T_{DC}$, Electromagnetic torque of DC machine (Nm); $T_r$, Pump Electromagnetic torque (Nm); $\omega_r$, Rotor electrical speed (rad/s).

## 1. Introduction

A nation's growth is significantly influenced by its sources of water, essential for domestic, drinking, large-scale irrigation, building, and electricity generation purposes. The demand for water amounts to approximately five liters per person per day [1–3], yet accessibility remains a challenge in many regions despite ample global water resources. This underscores the critical role of water pumps (WP) in transporting clean water from its source to areas of need, a task they have fulfilled for many years [4–6].

In contrast to fossil fuels, photovoltaic (PV) systems have gained attention for their environmental benefits and extensive development efforts aimed at enhancing efficiency and structure. Ref. [7] utilized the Newton-Raphson method to analyze PV cell equivalent circuit parameters, focusing on fill factor and open-circuit voltage using MATLAB-based software. Ref. [8] explored the modeling and simulation of an off-grid PV system incorporating an LC passive filter to mitigate harmonic components, achieving significant reduction in total harmonic distortion (THD) from 91.55% to 2.62% through MATLAB/Simulink simulations. PV systems have been deployed in various isolated locations for purposes such as irrigation, livestock watering, communal water supply, and rural electrification, offering advantages including quiet operation, minimal carbon emissions, and low maintenance costs [9, 10].

Electric drive systems constitute a substantial portion, approximately 70%, of global electricity consumption, with pumping systems alone consuming between 10% to 40% of total electrical energy produced [11, 12]. The shift towards renewable energy-based WP systems is steadily replacing diesel engine pumps due to their adverse environmental impact, high maintenance costs, and substantial initial investments [13, 14]. PVs are emerging as a viable solution in remote areas to meet daily energy needs, particularly for WP applications [13, 15, 16], offering sustainability and minimal maintenance costs without the added expense of fuel. These systems rely on stochastic changes in temperature and solar irradiation.

The induction motor (IM) plays a pivotal role in WP systems due to its affordability, simple mechanical design, and robust nature [17]. However, its nonlinear control characteristics present challenges. Indirect field-oriented control (FOC) is commonly employed to achieve decoupled flux and torque control, akin to separately excited DC motors, ensuring stable operation. Additionally, a model predictive speed control strategy for surface-mounted permanent magnet synchronous motors utilizing Laguerre functions integrates MPC with online quadratic programming for constrained optimization [18], demonstrating superior performance in steady-state and dynamic response compared to traditional methods.

Despite advancements, uncertainties and external disturbances can impact system efficiency [19, 20]. Independent PV-WP systems are widely deployed in rural areas for domestic, agricultural, and drinking water needs, eliminating the need for battery storage. Historically, WP demands were met by diesel-electric and fuel cell systems, which are environmentally polluting, require annual maintenance, and operate inconsistently. Power converters (PCs) serve as intermediaries between PV panels and load-driven motors, necessitating compatibility between motor characteristics and PV module specifications for efficient PV-WP operation. Numerous studies have documented drive-driven WP systems, encompassing both DC and AC drives [21–23].

For optimized PV power harvesting, various approaches have been developed [24–27]. The intermediate DC-DC switching PC plays a crucial role in two-step MPPT systems, facilitating PV module peak power point tracking. Alternatively, single-stage MPPT eliminates the need for DC-DC PC structures, maintaining PV module operation at optimal efficiency levels. Several DC-DC PCs, including SEPIC, ZETA, and CUK converters, have been utilized for PV MPPT applications [28–30]. The SEPIC converter, a fourth-order PC, enhances PV module

MPP optimization and motor WP system speed regulation, acting as an impedance matching interface between PV modules and PCs. Despite its benefits, the SEPIC converter exhibits pulsating current effects. Conversely, the ZETA converter offers simplified compensator circuitry and reduced output ripple compared to SEPIC, albeit with challenges related to flying capacitor size and buck-regulated power circuitry. The CUK converter maintains constant input/output current, enhancing transient response and reducing settling time compared to other DC-DC PCs. Ref. [31] evaluates PV-based SEPIC, CUK, Zeta, and Luo converters, highlighting the superior efficiency and reduced ripple under transient conditions offered by the Luo converter.

Furthermore, a Laguerre-based model predictive controller (LMPC) stabilizes and regulates voltage in a buck DC-DC converter [32], while a modified controller for a three-level, three-phase voltage source inverter employs Laguerre functions within model predictive control to enhance precision and reduce computational load, validated through experimental trials showing improved performance and reduced total harmonic distortion [33]. Nevertheless, the intermittent nature of power generation in such systems, without precise control over active and reactive power production, often poses challenges to the connected electrical systems, particularly in the context of IM-based WP systems. Consequently, the modeling and optimization of PV-WP systems with IMs have become a pivotal research domain aimed at enhancing system efficiency [34–36]. While the research landscape on IM control has witnessed prolific growth in recent years, with scientists worldwide continually publishing results, two predominant methods have solidified their position as the cornerstone of research in motor control: scalar control (SC) with open or closed loops and vector control (VC) or FOC. These techniques have served as the foundation for various derivative methods, particularly embraced by the variable-speed drive development industries, with direct torque control emerging as a notable example. It is essential to emphasize the advantages of VC over basic SC. VC offers effective machine control in both stable and transient operating conditions, making significant contributions to control objectives.

This study introduces an innovative approach to emulate PV water pumping systems, offering a cost-effective alternative for development and testing without the need for large-scale setups. By integrating a PV system emulator, boost converter with MPPT algorithm, and motor pump emulator into a unified framework, the system achieves efficient testing and development capabilities. The optimized scalar control strategy for motor pumps enhances dynamic performance and control precision, making the emulation setup highly effective.

The main contributions of this research are:

- Integrating PV and water pump emulators into a cohesive system validated through experimental setups using the dSPACE control desk DS1104, accurately reproducing solar panel characteristics.

- Introducing a novel MPPT algorithm based on Golden Section Optimization (GSO), marking its first application for tracking maximum power points under dynamic conditions and partial shading. This algorithm ensures rapid convergence and robustness against atmospheric variations, requiring no parameter tuning and relying solely on the PV module's open-circuit voltage. It outperforms traditional MPPT methods in real-world scenarios, enhancing global MPP tracking efficiency and simplifying implementation.

These innovations advance PV system emulation and MPPT techniques, providing practical solutions to enhance the efficiency and feasibility of solar energy systems.

The paper's structure includes a descriptive study of global photovoltaic pumping systems, exploration of control strategies for different system stages, detailed implementation of the proposed photovoltaic pumping system emulator with comparative performance analysis, and presentation of results and conclusions with key findings and implications.

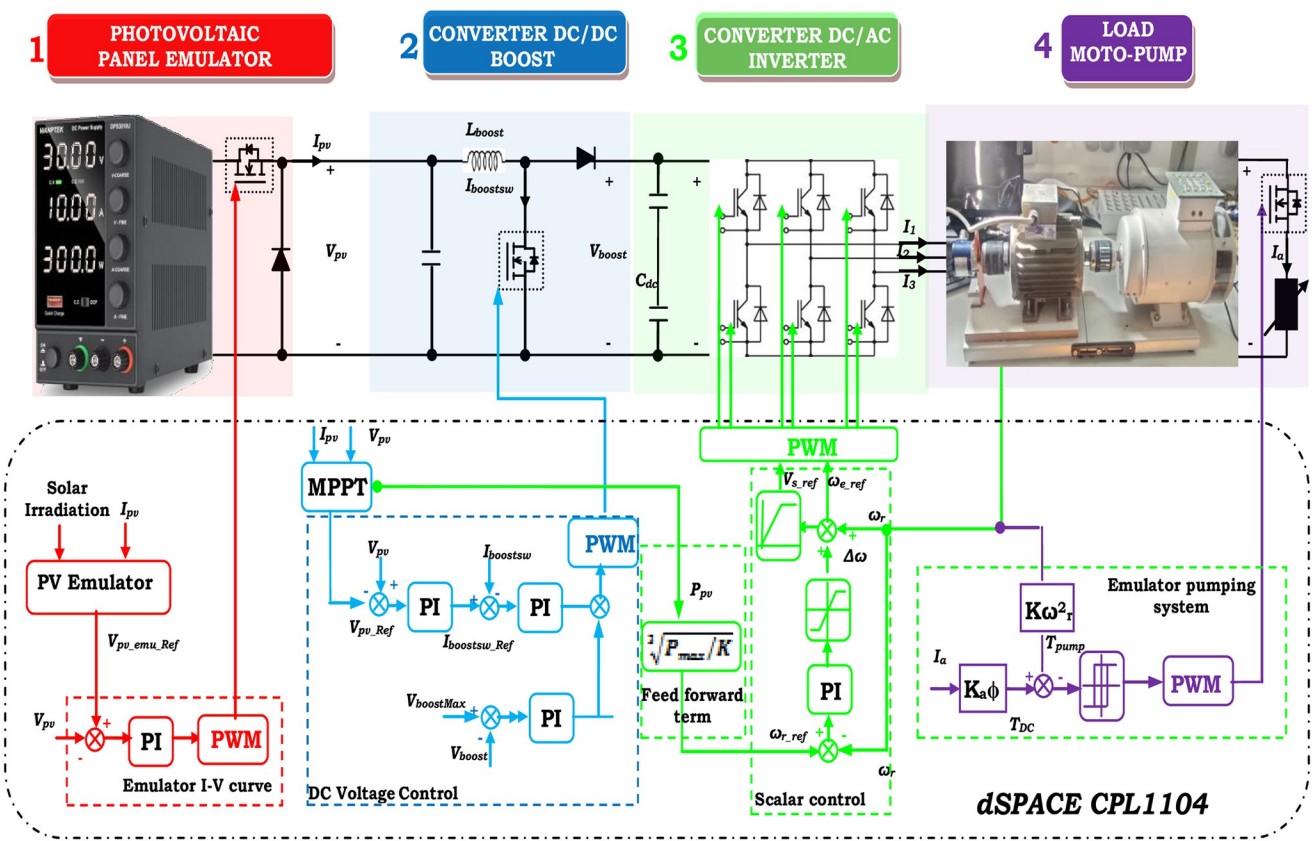

**Fig 1. Addressed system.**

## 2. Descriptive study of a global photovoltaic pumping system

The PV-WP system essentially includes a PV array, a DC/DC PC, an MPPT controller, a DC/AC PC, and a motor pump (MP), as shown in Fig 1.

The PV generator, commonly known as the PV array, plays a pivotal role in supplying electrical power to the system by efficiently converting PV energy into electrical energy [37]. In this study, a novel approach was employed to emulate the PV system, utilizing a DC power system in conjunction with a buck PC, all meticulously controlled by a DSPACE card. The second crucial component in the system involves a DC-DC PC, specifically a boost PC, governed by the MPPT technique. This MPPT control strategy ensures the optimum power output achievable from the PV array emulator (PVAE). The third element of the setup is the DC-AC PC, commonly known as an inverter. Its primary function is to convert the DC into AC voltage, facilitating the supply of power to the AC motor integrated into the MP system. The MP system comprises an IM paired with a CP, with the selection of the IM tailored to meet the power requirements of the pump.

## 3. Control strategies

### 3.1. Control of PVAE system

In this part, the output current of the panel and the irradiation level are utilized as inputs to generate an output voltage. The P-V electrical physical appearance of the PVAE should be similar to the characteristics of the PV array simulated and demonstrated in Figs 2 and 3. The

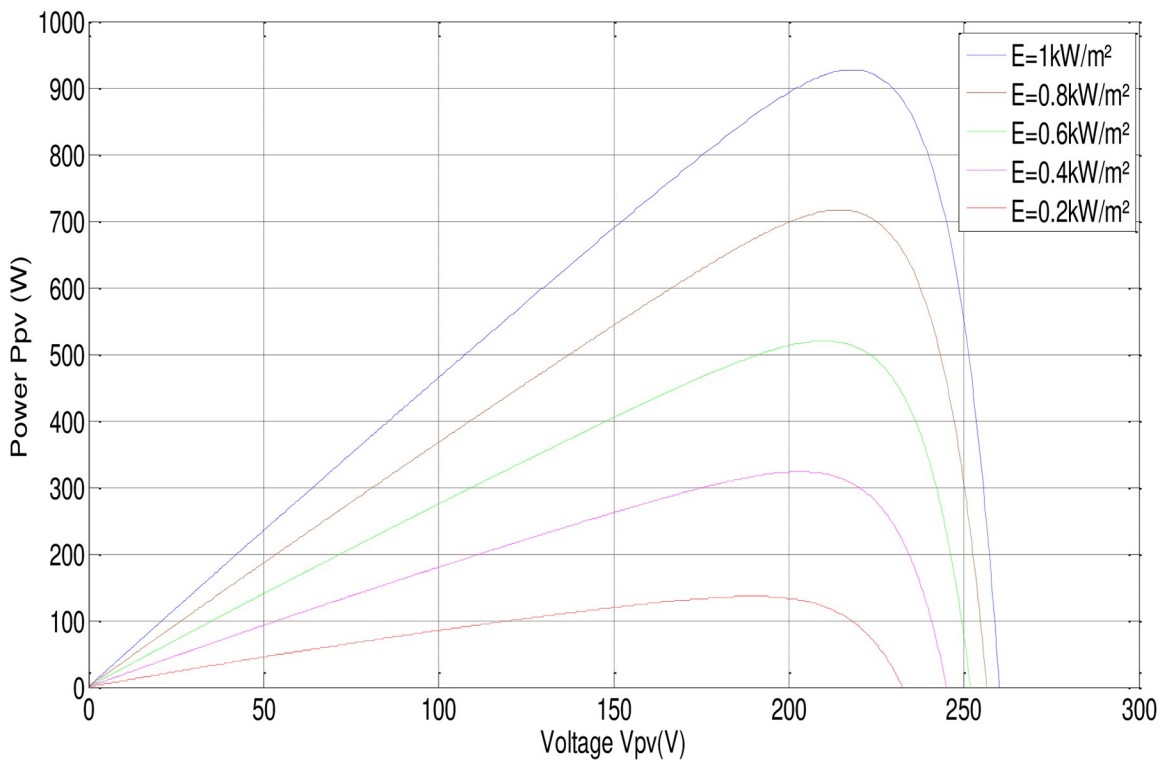

(a)

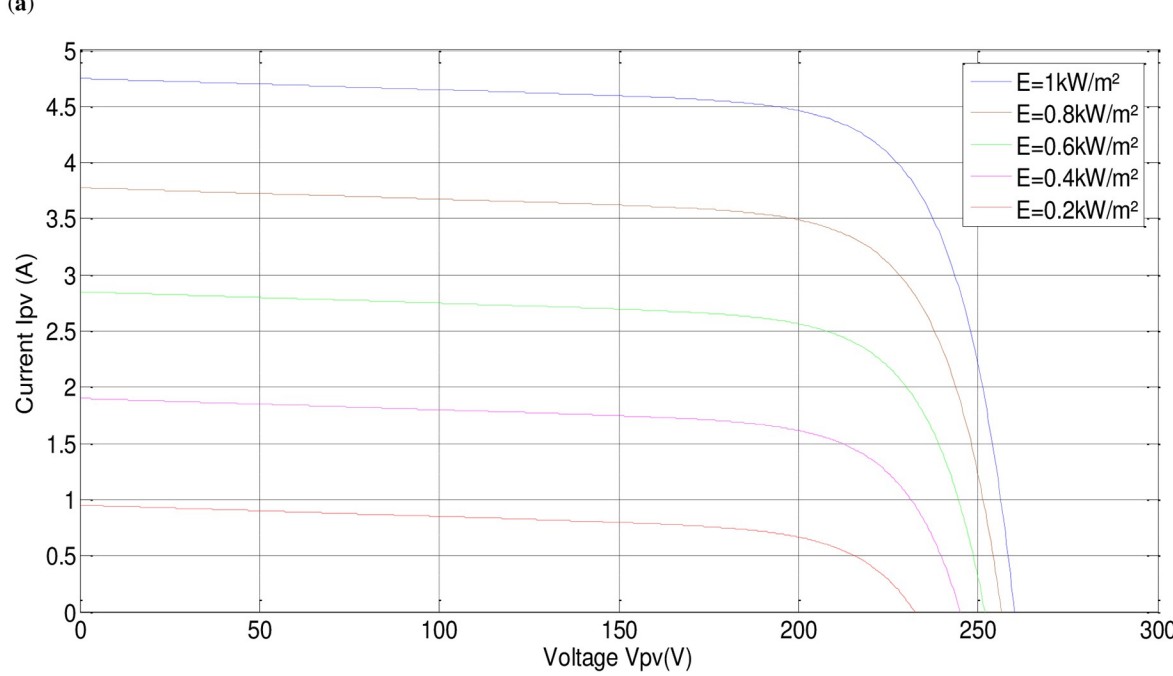

(b)

**Fig 2. P-V, I-V characteristics with fixed temperature, (T = 25˚C).**

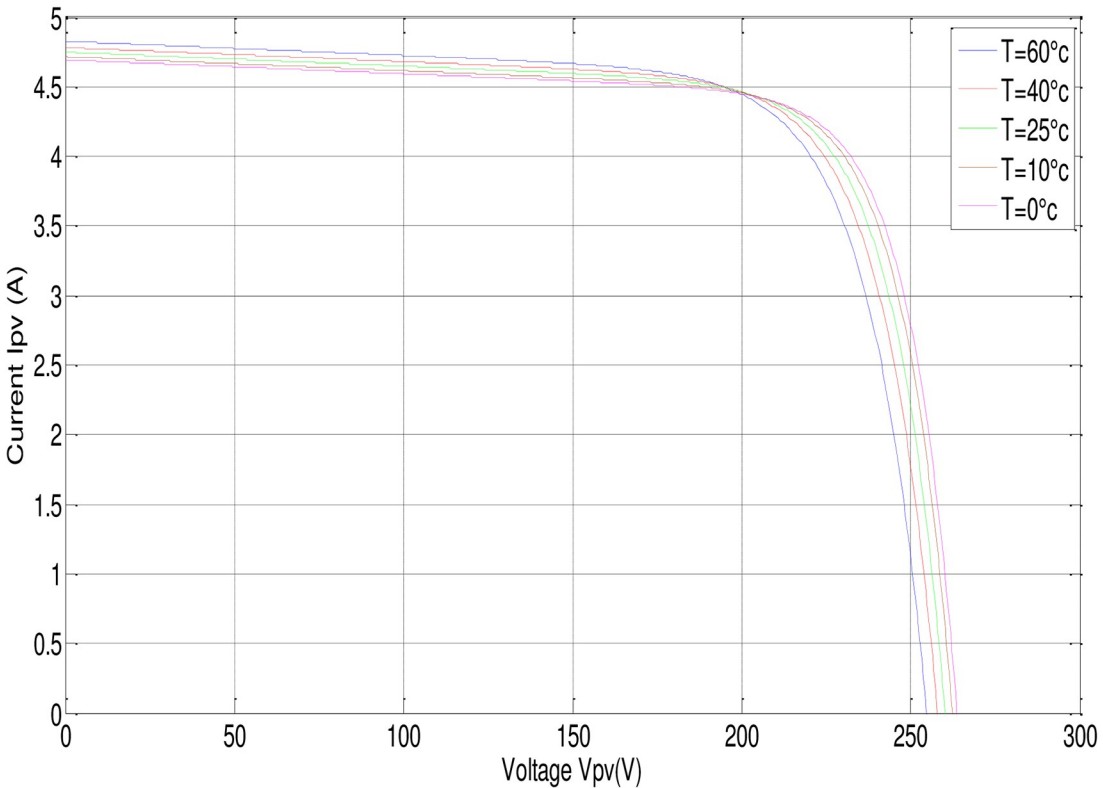

(a)

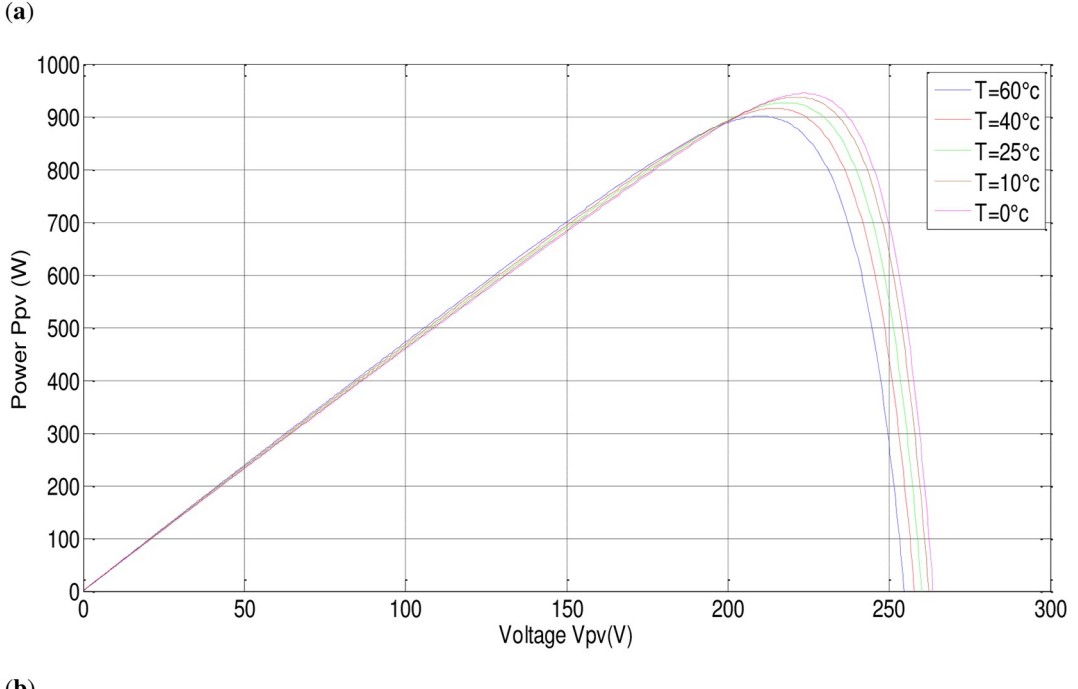

(b)

**Fig 3. P-V, I-V characteristics with fixed irradiance, (G = 1000W/m$^2$).**

**Table 1. PV panel characteristics.**

| PV panel (BP SX150S PV) | Values |
|---|---|
| Maximum power ($P_{max}$) | 150 (W) |
| Voltage at $P_{max}$ ($V_{mpp}$) | 34.5 (V) |
| current at $P_{max}$ ($I_{mpp}$) | 4.34 (A) |
| Warranted minimum ($P_{max}$) | 140 (W) |
| Open circuit voltage ($V_{oc}$) | 43.5 (V) |
| Short circuit current ($I_{sc}$) | 4.75 (A) |
| Temperature coefficient of ($I_{sc}$) | (0.065±0.015)(%/˚c) |
| Temperature coefficient of ($V_{oc}$) | (160 ±20) (mV/˚c) |
| Temperature coefficient of power | (0.5±0.05) (%/˚c) |
| NOCT | 47±2 (˚c) |

PVAE emulated the source of approximately 1 (kW), so six modules have been used in series, sufficient to supply the IM. Simulation using MATLAB/SIMULINK software of four modules connected in series with the parameters given in Table 1 gives the following results:

The process of determining the operating point given the load, irradiance, and temperature is known as the PVAE control approach. To become PVAE, it combines the PC and PV models. The control approach has an impact on the PVAE's different performance. A virtuous control method has minimal processing overhead, resilience, high adaptability in simulating different PV models, low processing burden, and correct output voltage and current similar to the PV model. It is also simple to implement. When implementing PVAE, a variety of control mechanisms are employed. Because of its ease of use, the direct referencing method (DRM) is frequently employed in the PVAE [32, 38]. Fig 4 presents the design of the PVAE based on the DRM. The equivalent circuit above deduces that [10, 29, 30, 33, 39].

$$I = I_{ph} - I_D - I_{sh} \tag{1}$$

$$I = I_{ph} - I_0 \left[ exp \left( q \frac{(V + R_S.I)}{N_S nKT} \right) - 1 \right] - \left( \frac{V + IR_s}{R_{sh}} \right) \tag{2}$$

To achieve the desired power output, it is essential to connect the required number of PV modules both in series and in parallel. This arrangement is commonly referred to as a PV array [10, 33].

$$I_{array} = N_p I_{ph} - N_p I_0 \left[ e^{\frac{q\left( \frac{V_{array}}{N_S} + \frac{R_S I_{array}}{N_p} \right)}{nKT}} - 1 \right] - \frac{V\left( \frac{N_p}{N_S} \right) + I_{array} R_S}{R_p} \tag{3}$$

A current model for the PV array (Eqs (2) and (3)) is deployed to compute the reference voltage. $V_{PV-array-ref}$, considering irradiation variations. $V_{PV-array-ref}$ is compared to the buck PC's output voltage and regulated using a PI regulator to generate the PWM for the buck PC, simulating the PV panel's behavior.

## 3.2. Control strategy of converters

There are two parts to the system control as depicted in Fig 5. Using one of the MPPT approaches, the PV array's maximum power is tracked. The duty cycle of the dc-dc boost PC is

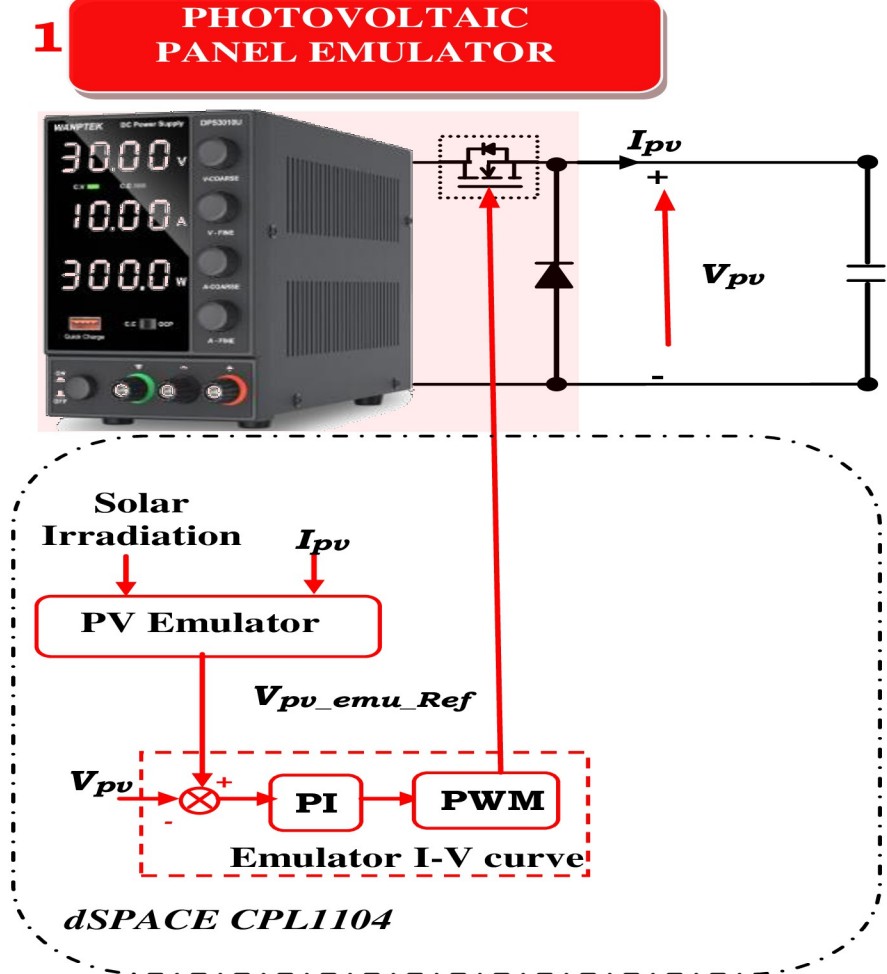

**Fig 4. PV emulator and boost converter scheme.**

the stage's control variable. The second part is the recommended linear V/f control, which uses conventional sinusoidal PWM to produce the logic switching signals [40].

Two PI control loops are used to maintain the DC-DC boost PC's inductor current constant at a preset standard value. In this instance, the MPPT's methods are used for measuring the panel output voltage and current to produce a standard voltage ($V_{PV\text{-}ref}$) that corresponds to the maximum power voltage.

In this study, a new MPPT algorithm named "Golden Section Search (GSS)" was simulated and compared with conventional ones (P&O and Incremental). This new algorithm operates through a series of narrowing steps aimed at identifying the range of values containing the extremum value. This process, known as the GSS [41], involves evaluating function f(x) at two points, $x_1$ and $x_2$, selected within the interval [a, b]. These points divide the interval into two halves, ensuring that the ratio of the lengths of the larger and smaller subintervals to the full interval length remains constant. Consider a line segment [0, 1] as depicted in Fig 5. Then:

$1/r = \frac{r}{1-r}$ i.e. $r^2 + r - 1 = 0$ hence r = 0.618

$x_1 = b - r(b - a)$ i.e. $x_1$ is 0.618 of interval away from b(1)

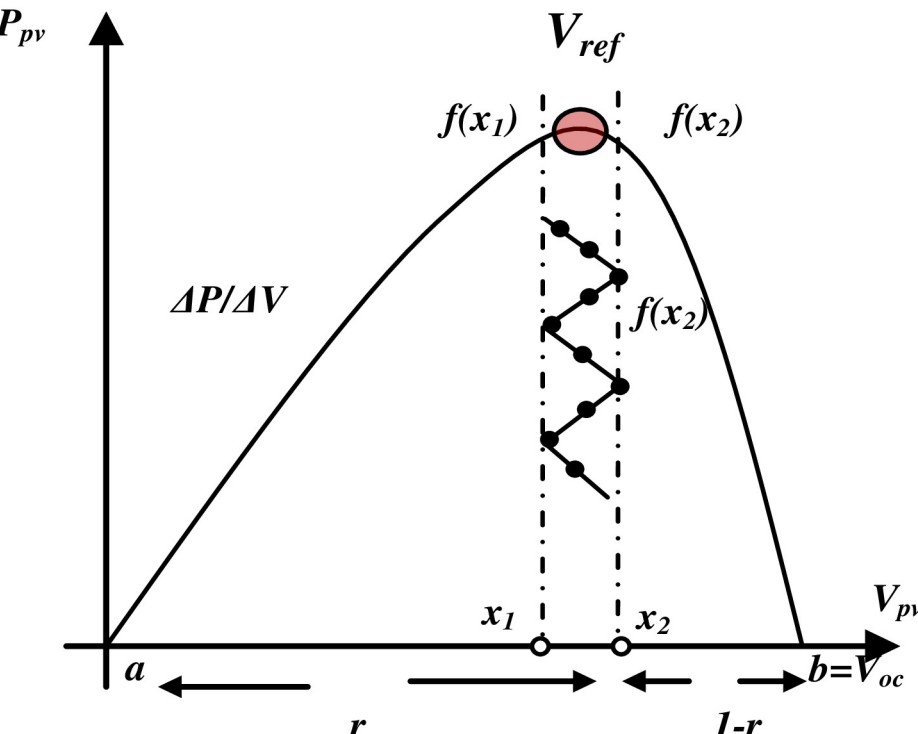

**Fig 5. The division of interval and the interval markings on the characteristics PV.**

$x_2 = a + r(b - a)$ i.e. $x_2$ is 0.618 of interval away from $a(2)$

The P-V characteristics are the operating characteristics of a photovoltaic system that uses GSS for maximum power point tracking. Here, f(x) represents the power, whose maximum value needs to be recorded, and $x_1$ and $x_2$ represent the array voltage. As seen in Fig 5, the range of operation is from zero to open circuit voltage ($V_{oc}$), while a = 0 and b = $V_{oc}$. Fig 6 displays the flowchart for utilizing GSS to discover a function's maxima [42].

As depicted in Fig 7, the reference voltage ($V_{PV-ref}$) is subtracted from the panel voltage, producing an error that serves as the reference current ($I_{boost\ SW-ref}$) as it passes through a PI controller. The boost converter's switching duty cycle is then obtained by subtracting a periodic sample of the boost converter's switching current ($I_{boost\ SW}$) from the reference current. This error is also passed through a PI controller. Thus, the panel voltage is indirectly controlled by the switch's current control. The panel voltage must drop as the current reference rises. Consequently, the signals used to calculate the voltage error differ from those used to calculate the current error.

There is just one energy-storing component in stage 3 (DC link capacitor ($C_{dc}$). Under transitory circumstances, considering the lossless operation of the system:

$$P_{pv} = P_{Cdc} + P_r$$
$$P_{Cdc} = V_{boost} C_{dc} \frac{dV_{boost}}{dt}$$

$$(4)$$

The symbols $P_{pv}$, $P_{Cdc}$, and $P_r$ are the PV, $C_{dc}$, and motor output power.

$$P_{pv} = V_{boost} C_{dc} \frac{dV_{boost}}{dt} + T_r \omega_r, \left( T_r = T_{pump} \right)$$

$$(5)$$

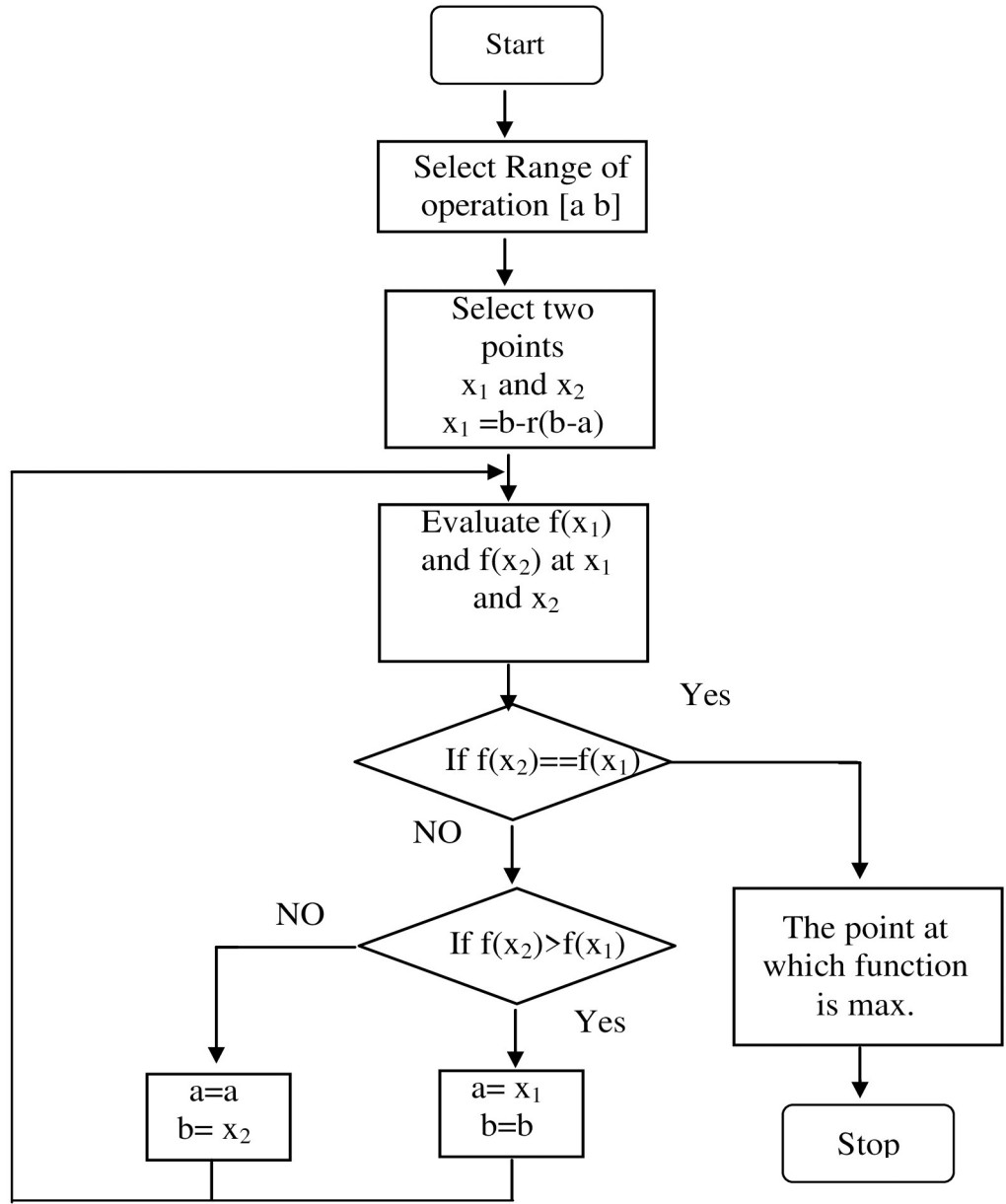

**Fig 6. Flowchart of the proposed GSS algorithm.**

During steady-state conditions, $V_{boost}$ remains constant, so,

$$\frac{dV_{boost}}{dt} = 0 \tag{6}$$

Hence (16) becomes:

$$P_{pv} = T_{pump}\, \omega_r \tag{7}$$

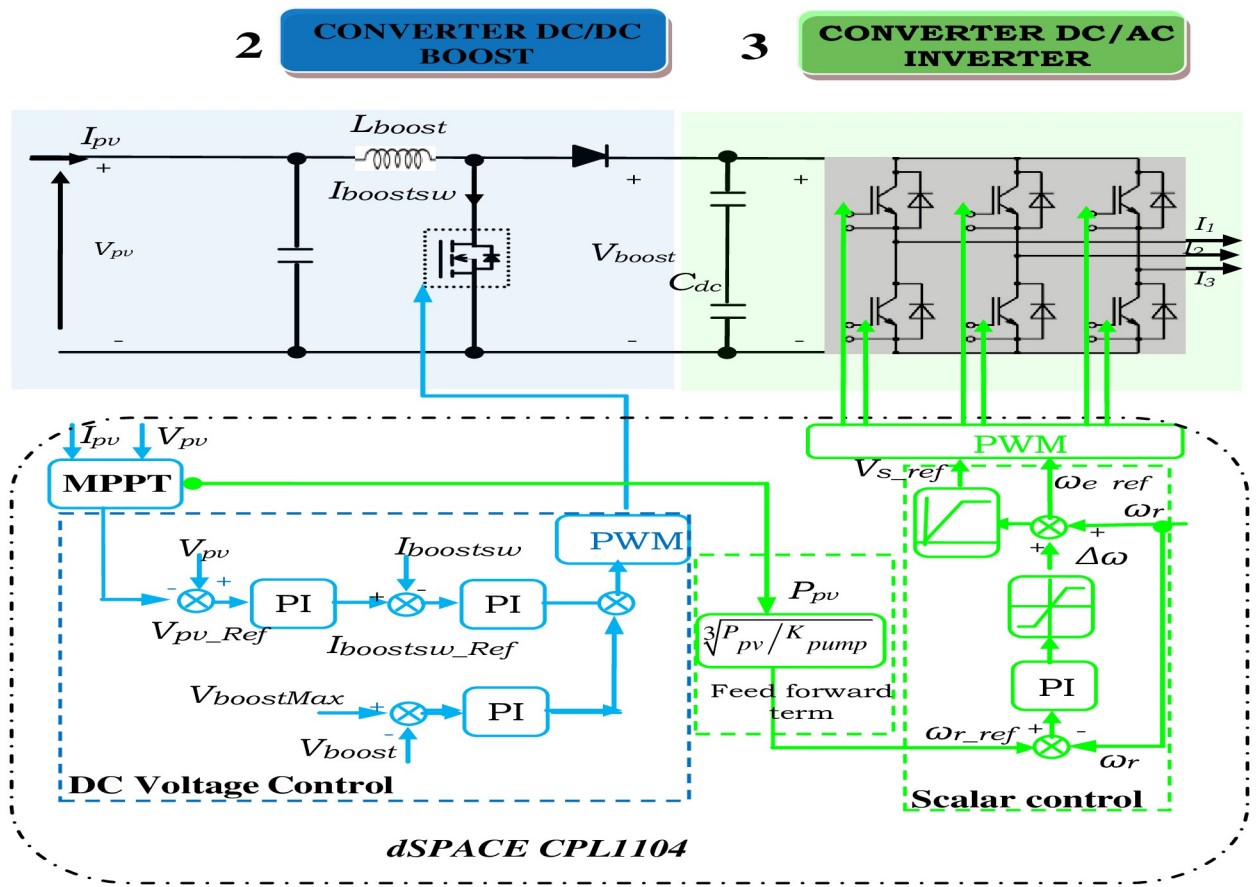

**Fig 7. PV emulator and boost converter scheme.**

Knowing that for the MP, the load torque is expressed as a function of the speed, as follows:

$$T_{pump} = K_{pump}\,\omega_r^2 \tag{8}$$

Finally,

$$P_{pv} = K_{pump}\,\omega_r^3 \Rightarrow \omega_r = \sqrt[3]{P_{pv}/K_{pump}} \tag{9}$$

Using (9), one can calculate the IM's speed reference based on the PV array's output power. The reference speed and the DC-link voltage (Vdc) control loop's output are then used to calculate this reference frequency. By employing sinusoidal PWM on the output from V/f control, the switching logic for the voltage source PC is produced.

The computed reference frequency from (10) will be greater than the accurate value as $P_{pv} > P_L$ and motor slip cannot be zero, especially when the load is at its maximum. By employing a different PI controller, the difference can be adjusted by deducting a compensatory term ($\Delta\omega$) that is derived from the feed-forward term and the steady state condition of the Vdc as expressed in the equations from 4–10. To maintain a power balance over the $C_{dc}$, the term ($\Delta\omega$) compensates for the motor losses. Therefore, using (10), it is possible to finally compute the necessary motor reference speed.

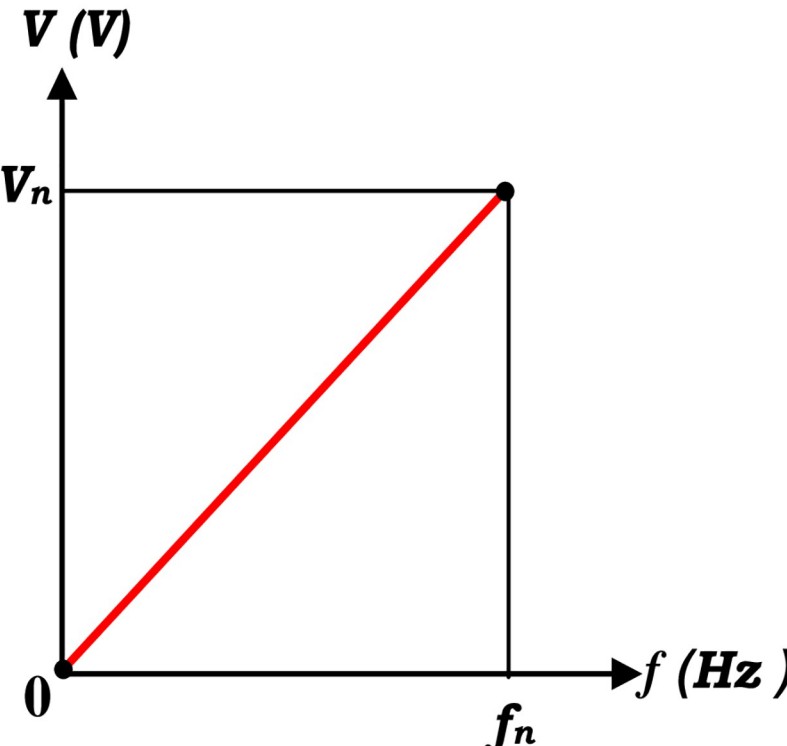

**Fig 8. The relationship in a linear V/f control between the frequency and stator voltage.**

Next, using this reference speed and the supposition of a negligible motor slip, the needed desired operating frequency is estimated from (10):

$$\Delta\omega = \left( k_p + k_i \int dt \right) * \left( \omega_{r\_ref} - \omega_r \right) \tag{10}$$

The amplitude of the stator flux in linear V/f control is inversely related to frequency ($\phi \propto$ V/f) and proportional to stator voltage as illustrated in Fig 8. Therefore, for linear V/f-based supervision, maintaining a constant (V/f) ratio will result in constant air gap flux amplitude.

The reference frequency is the only input to the indicated linear (V/f) control, which is then utilized to produce the necessary frequency modulation index (the ratio of the frequencies of the sinusoidal reference signals to that of the "triangular" carrier signal) of the three sinusoidal voltage references. At last, the voltage source inverter's switching signals are generated via conventional SPWM.

### 3.3. Control of the moto-pump system

The pumping system, depicted in Fig 9, features an independently excited DC generator coupled to an IM, emulating pump system behavior. In addition, the rated values of the employed machines are listed in Table 2.

The independently excited DC generator torque is given by the following equation:

$$T_{DC} = K_a \phi I_a \tag{11}$$

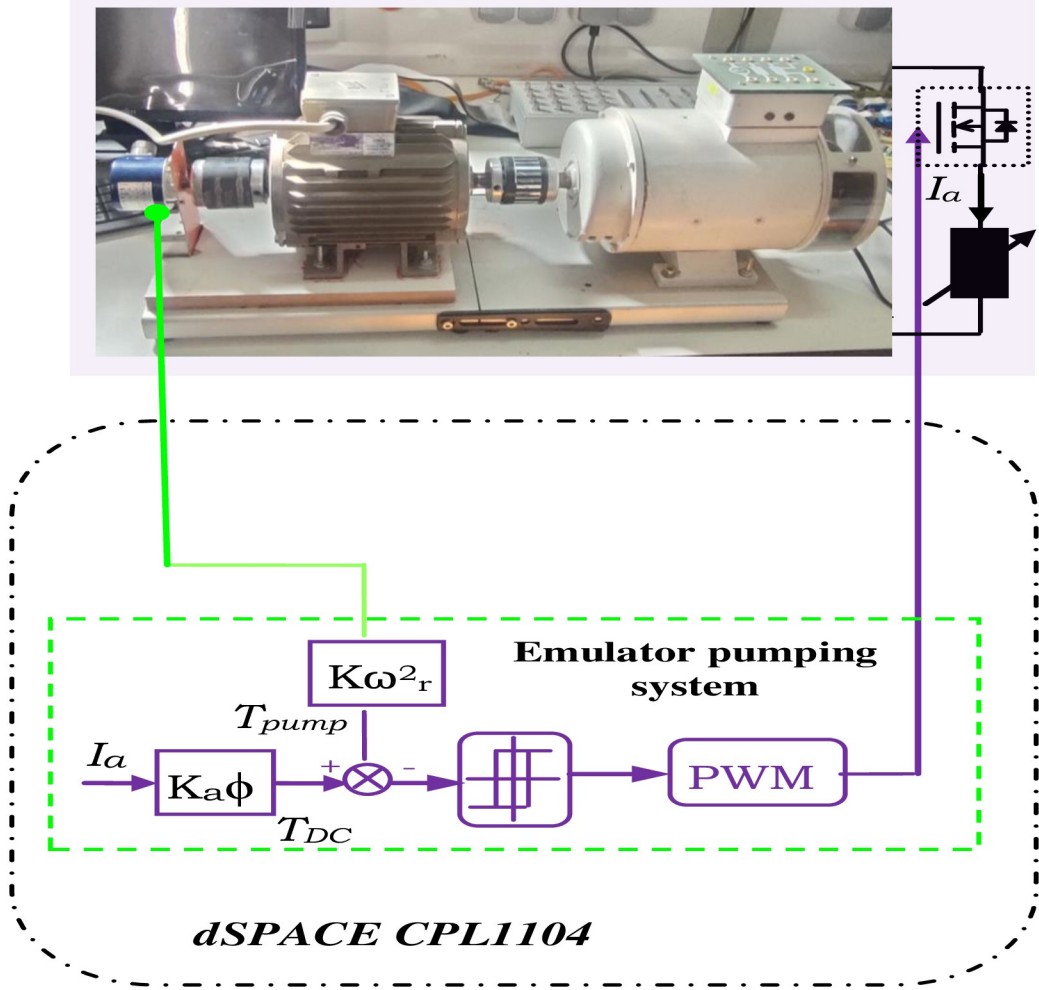

**Fig 9. Motor pump control scheme.**

**Table 2. The rated values of the IM and separately-excited DC generator.**

| Induction machines | Power | 0.75 | kW |
|---|---|---|---|
| | Frequency | 50 | Hz |
| | Voltage (Δ/Y) | 220/380 | V |
| | Current (Δ/Y) | 3..16/1.83 | A |
| | Speed | 2850 | rpm |
| | Pole pair (p) | 2 | |
| Separately-Excited DC generator | Power | 1 | kW |
| | Voltage | 220 | V |
| | Rated Current | 6 | A |
| | Excitation Current | 0.53 | A |
| | Speed | 1500 | rpm |

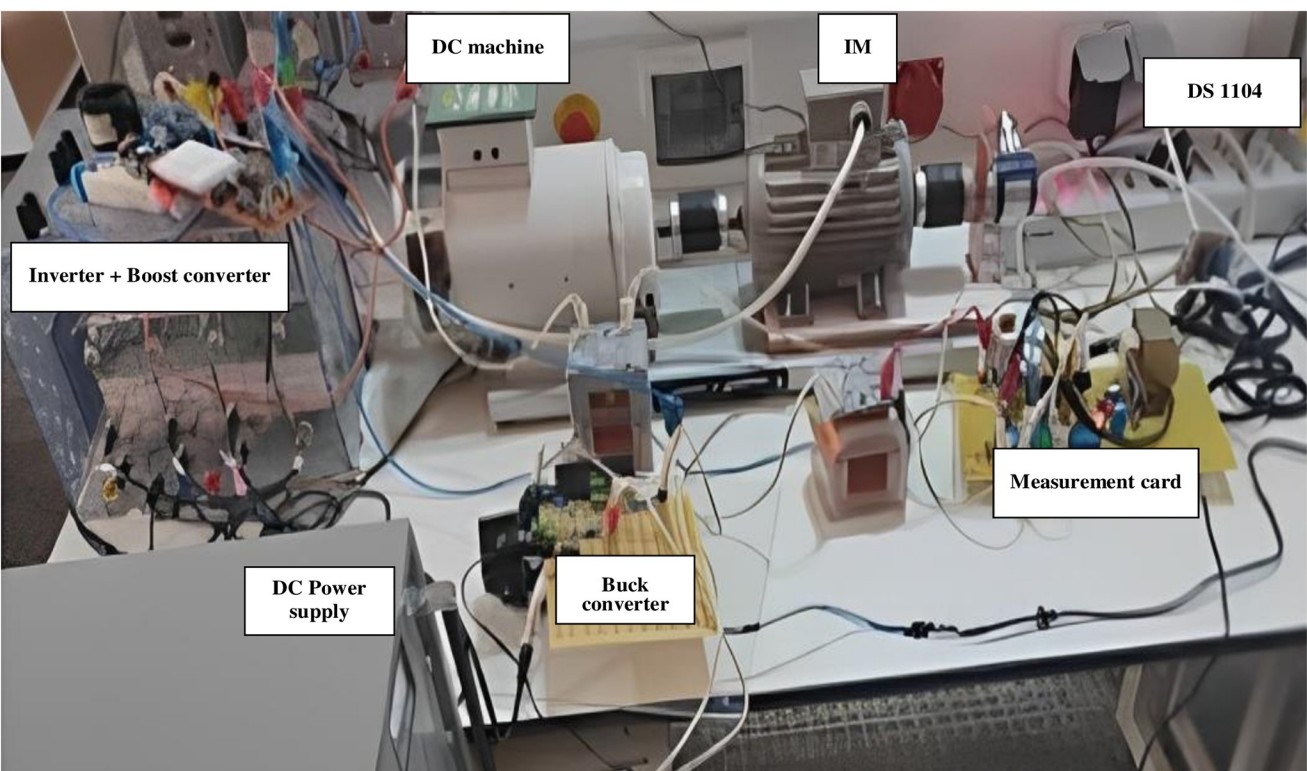

**Fig 10. Implementation of PV-WP system emulation via DS1104 hardware.**

The characteristic torque-speed relationship of the pump load reveals that load torque is proportional to the square of motor speed ($\omega_r$) presented in Eq (8). Emulating the pumping system involves subtracting the DC generator torque from the instantaneous pumping torque, resulting in an error that is controlled to replicate centrifugal pumping behavior.

## 4. Implementation of proposed photovoltaic pumping system emulator

### 4.1. Description of the laboratory setup

According to Fig 10, the control strategies are executed on the dSPACE 1104 platform, which oversaw the PV emulator, boost converter, and inverter for the WP system. The implementation of controllers is conducted within Simulink. Using the C code generator, MATLAB's Real-time Workshop programs the dSPACE ACE kit 1104 CLP module. The controllers are simulated in MATLAB and then loaded onto a dSPACE controller module. The real-time interface (RTI) of MATLAB facilitates the graphical configuration of controller inputs and outputs in Simulink.

### 4.2. Simulation and *experimental results and discussion*

**4.2.1. Simulation results.** MATLAB/Simulink software is employed for conducting various tests to evaluate the success and robustness of the suggested control. In Fig 11, a rapid variation in irradiation is initially applied, starting at 0.8 kW/m$^2$ of sunlight. At 0.6 seconds, it is

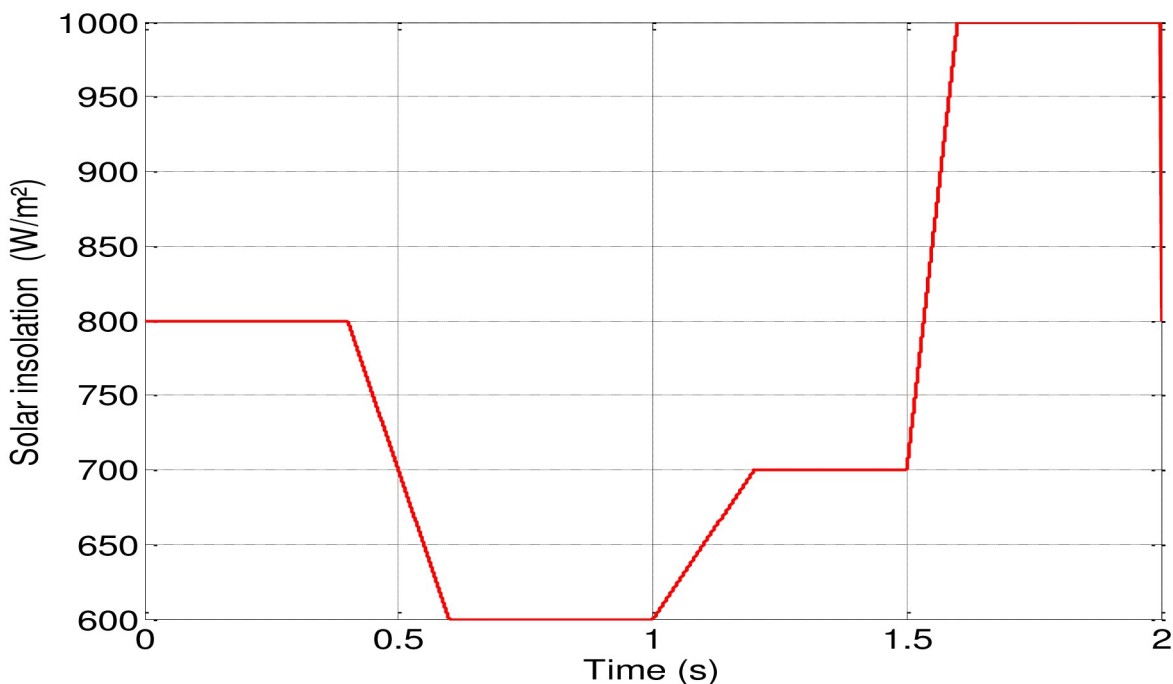

**Fig 11. Profile of the applied solar insolation.**

adjusted to 0.6 kW/m$^2$, followed by an increase to 0.7 kW/m$^2$ at 1.2 seconds, and further to 1 kW/m$^2$ at 1.6 seconds, while maintaining a temperature of 25°C.

Figs 12 and 13 depict how changes in insolation affect the motor-pump characteristics and illustrate the DC link voltage and speed response.

Fig 14 presents the PV array's power response using three different MPPT techniques. The INC-controlled PV system achieves steady-state with higher power values more quickly compared to systems employing P&O and GSS algorithms.

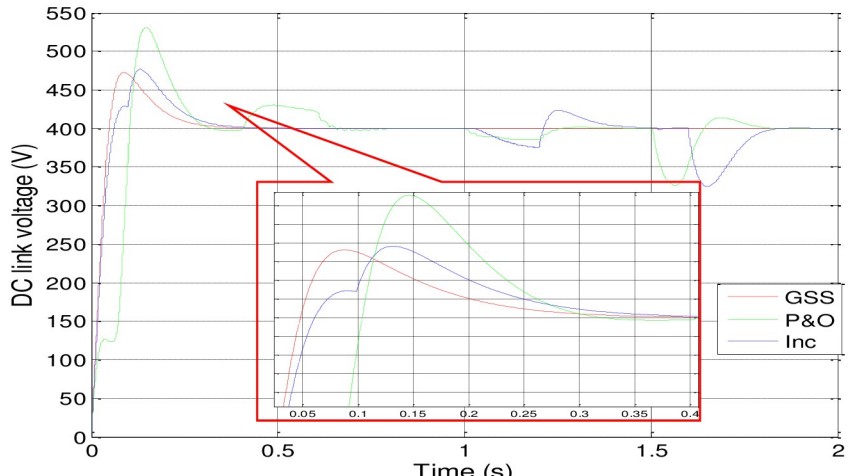

**Fig 12. Response of the DC link voltage.**

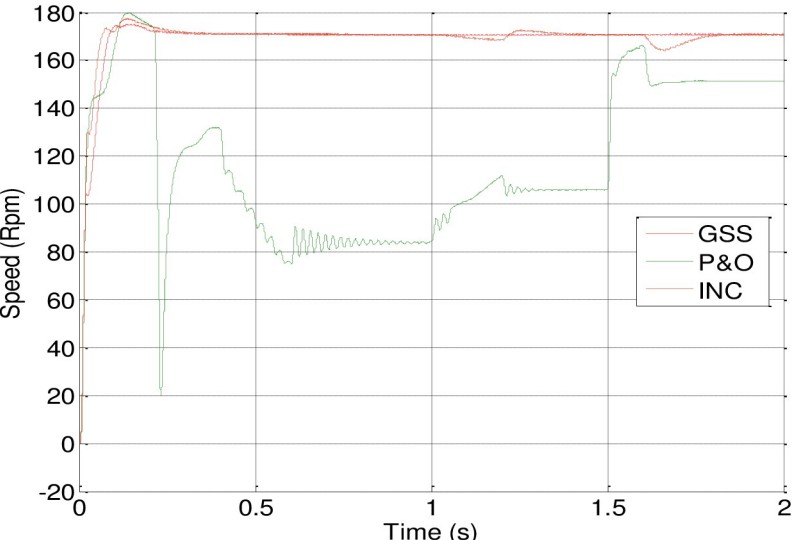

**Fig 13. Response of motor-pump; speed.**

Also, the GSS and INC MPPT techniques with the P&O-controlled are used in PV system to make the PV generator to operate at its maximum power point in spite of variations in the weather. In comparison to the other two, the GSS control operates more quickly. The GSS and INC MPPT techniques force the PV generator to operate at its maximum power point in spite of variations in the weather.

**4.2.2 Comparative studies.** Table 3 and the graphic summarize and exhibit the differences in the performance of the three techniques throughout the simulation.

According to the Table 3, it appears that the GSS technique significantly improves system stability. The DC link voltage and speed remain in a steady state without disturbance during variations in solar irradiation, except during the initial transient period, where it exhibits some overshoot, although less than the P&O or INC techniques, despite being somewhat delayed.

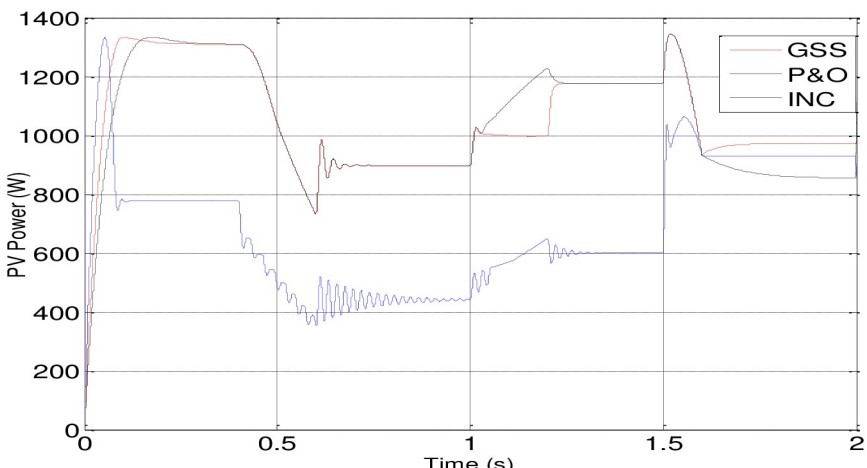

**Fig 14. Response of the PV power.**

Table 3. Comparative study of the techniques' performance.

| Profile transient time | Overshoot / Undershoot (%) | | | | | |
|---|---|---|---|---|---|---|
| | DC link voltage response | | | Speed response | | |
| | P&O | INC | GSS | P&O | INC | GSS |
| 0–0.3 | 32.5 | 18.75 | 17.5 | 5.88 | 3.2 | 1.7 |
| 0.3–1 | 7.5 | 1.25 | 0 | -23.53 | 0 | 0 |
| 1–1.2 | 3.75 | -6.25 | 0 | -35.29 | -1.18 | 0 |
| 1.2–1.5 | 1.25 | 6.25 | 0 | -38.23 | 1.18 | 0 |
| 1.5–1.6 | -18.75 | 1 | 0 | -11.76 | -3.53 | 0 |
| 1.6–2 | 3.5 | -18.75 | 0 | -11.76 | 0 | 0 |

However, the P&O and INC techniques display disturbances during insolation variations, with the P&O technique being particularly more affected.

This graph exhibit the most efficient method to extract maximum power from the PV array. It seem that GSS method is more efficient at any isolation changing. Dispite that the INC is also seems like GSS but at the last transient time it loose its performance (Fig 15).

**4.2.3. Experimental results.** *(A) Steady-state performance of PV emulator.* To validate the performance of control techniques, the PV emulator is assessed under varying irradiation levels, as illustrated in Fig 16. Initially, the P&O MPPT method is employed on the I-V and P-V curves derived from the panel emulator (Fig 2). As illustrated in Fig 17, it showcases accurate maximum power tracking across a spectrum of irradiation levels, ranging from 1000 (W/m$^2$) to 500 (W/m$^2$). As can be seen in Fig 18, the emulator successfully produces voltage ($V_{pv}$) and current ($I_{pv}$) responses that closely align with each irradiance variation. Furthermore, the DC bus voltage ($V_{boost}$) demonstrates effective tracking when compared to the $V_{pv-ref}$, which corresponds to the MPP.

*(B) Steady-state performance of the load (MP).* The delay in reaching the peak value of the PVAE's power output was observed, impacting the response of the system to load

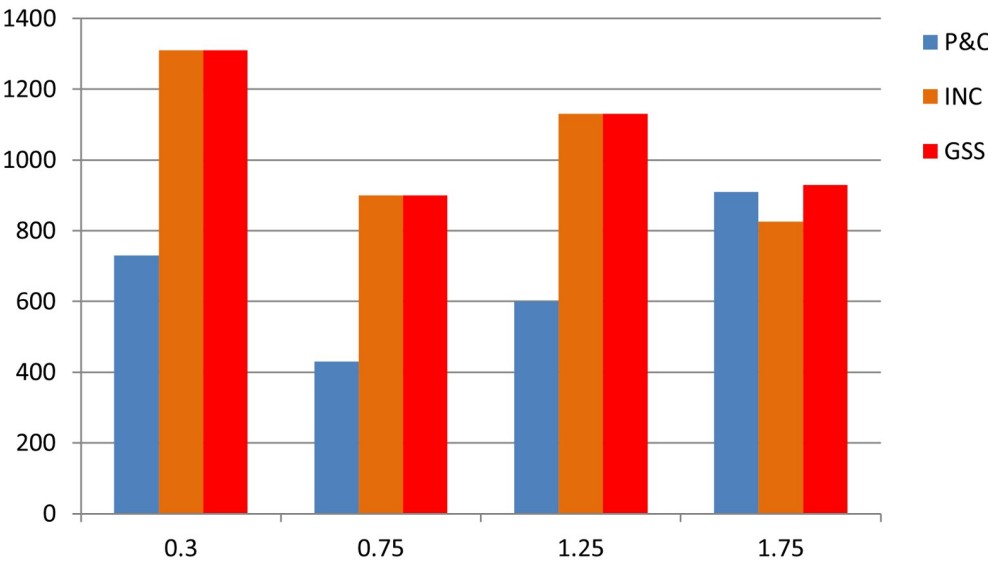

**Fig 15. Graph 1.** Power fluctuation of the three techniques.

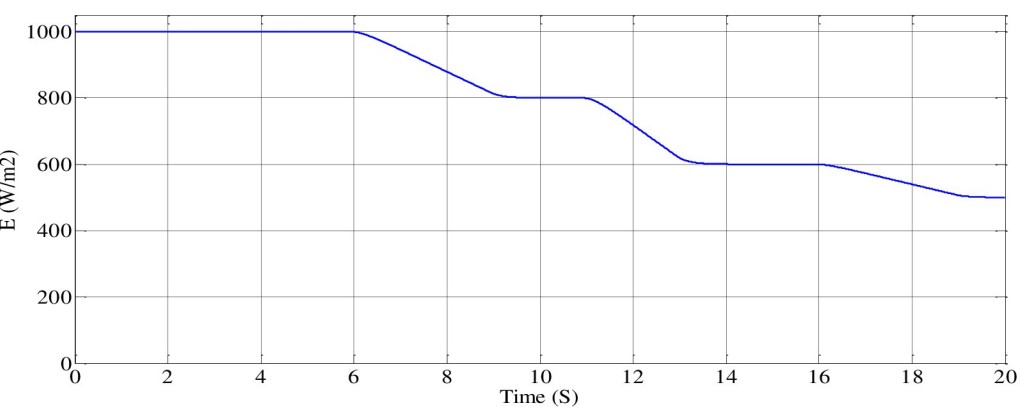

**Fig 16. Profile of irradiance variation curve.**

demands. Adjustments have been made to ensure effective response of the emulator to the MPPT reference voltage, as illustrated in Fig 19. Initially, power is received by the DC link of the inverter. When the boost PC's switching device is in the off position, the voltage across the DC link terminals of the inverter matches the open-circuit voltage of the PV

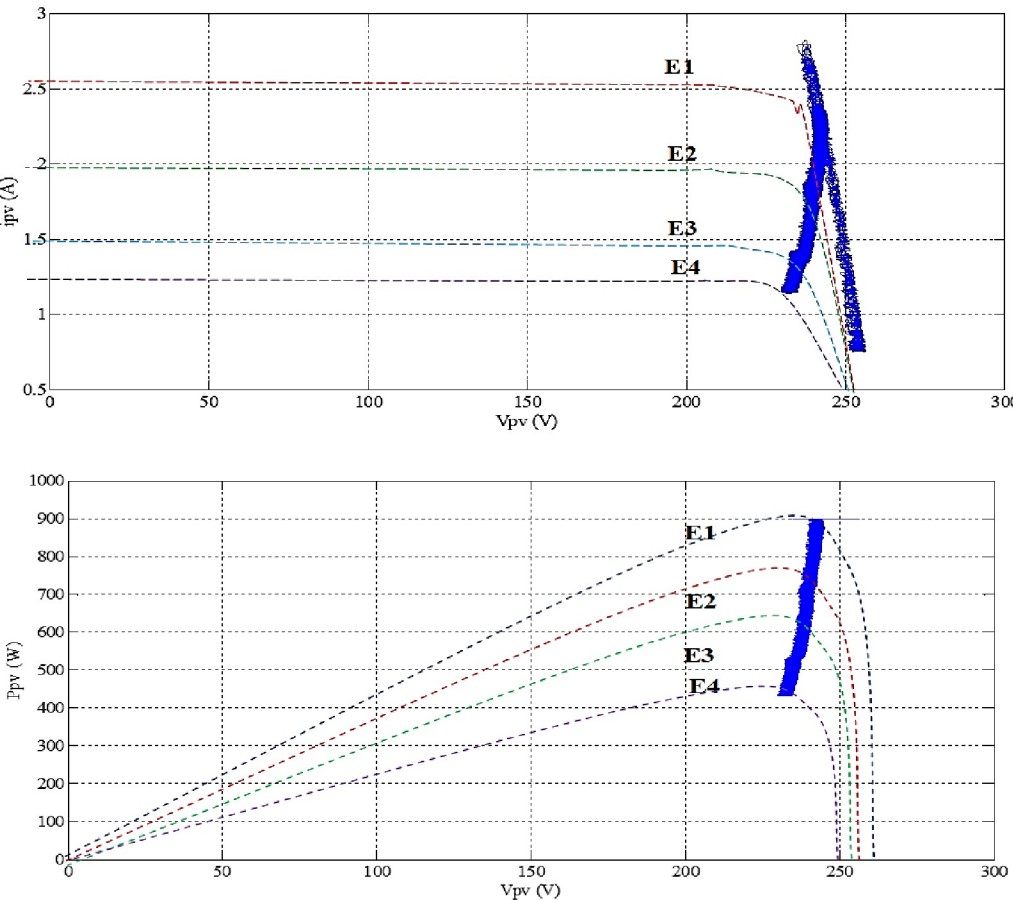

**Fig 17. I-V and P-V characteristics of the emulator during irradiation changing.**

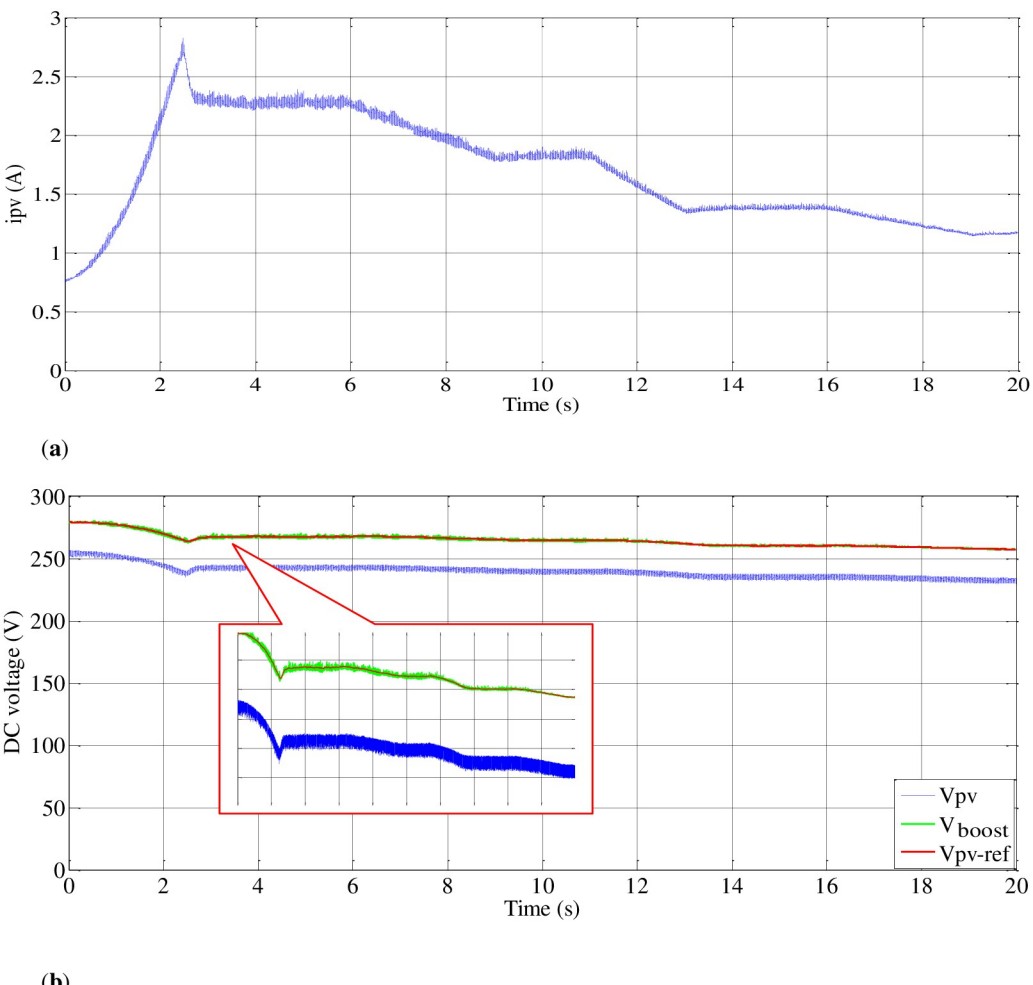

**Fig 18. PV emulator variation current and boosted voltage during irradiance variation.**

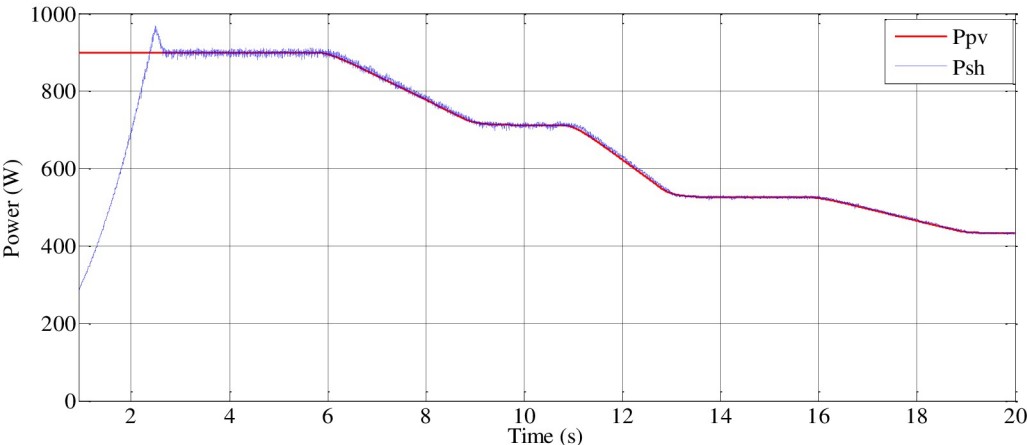

**Fig 19. PV emulator and shaft power variation.**

generator, gradually decreasing as motor speed increases. Simultaneously, the current of the PVAE begins at zero and progressively rises to the maximum current (Imax), as depicted in Fig 18.

At t = 2.35 (s), activation of the boost PC occurs, and the system converges to its corresponding MPP. Adjustment of the intermediate circuit voltage to the reference value is achieved through the action of a PI regulator. Fig 18 verifies that the motor current consistently remains below its rated current due to a smooth startup procedure designed to extend the motor's lifespan. Validation of this approach is supported by emulator panel curves, confirming consistent operation of the system at peak power with control of the Boost converter output voltage while accommodating the load connected to the inverter output.

As depicted in section 4 (Fig 10), which focuses on the pumping system, an induction motor is coupled with an independently excited direct current generator to replicate the behavior of the pumping system. Fig 20 illustrates that the pumping torque is meticulously regulated to achieve a centrifugal pumping behavior that closely aligns with the DC generator's torque profile.

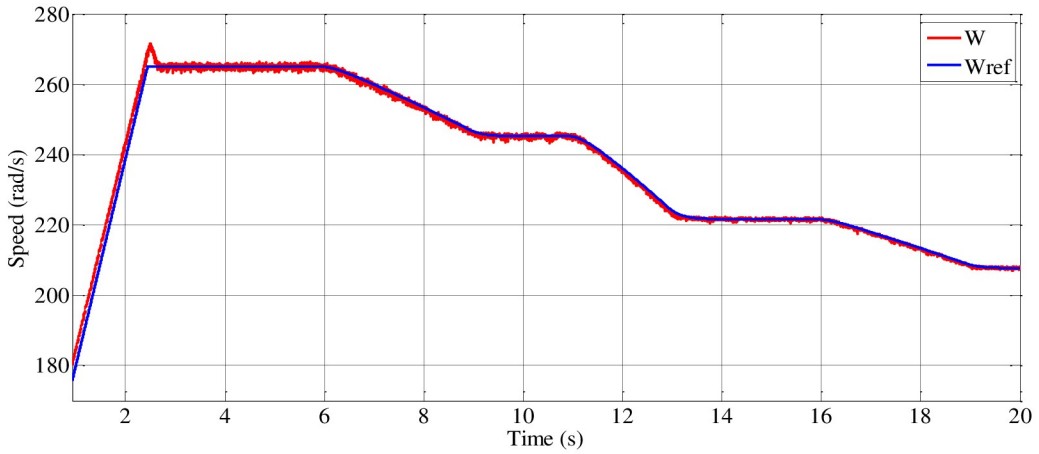

**(a)**

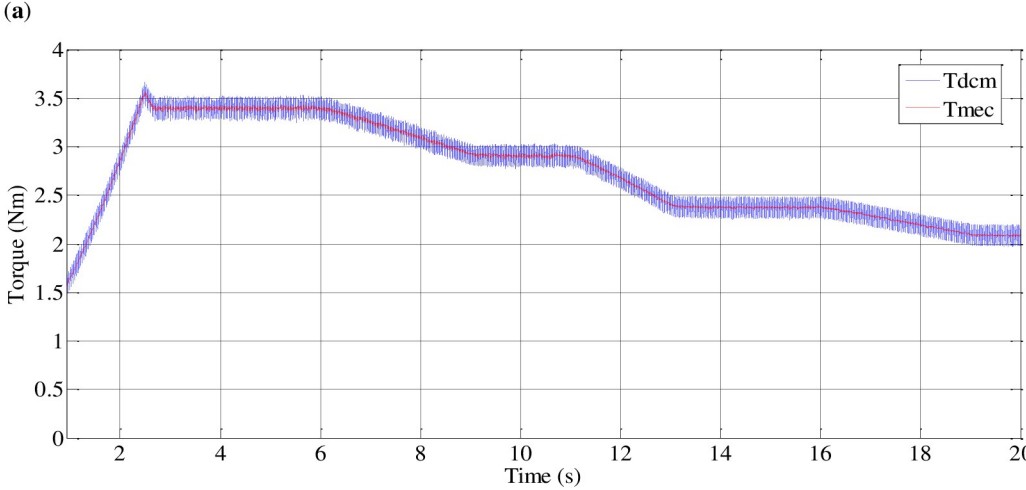

**(b)**

**Fig 20. Speed and torque curves during irradiation changing.**

## 5. Conclusions

The proposed system integrates a photovoltaic system emulator, a boost converter with an MPPT algorithm, and a motor pump emulator, effectively combining PV and WP emulation while optimizing system dynamics. The innovation lies in the fusion of these emulators into a single chain, allowing for precise control and faithful reproduction of real solar panel characteristics. The main objective of the study is to develop an adequate emulator for PV water pumping systems and evaluate an improved control algorithm. A proposed control using an MPPT based on both conventional (P&O and Inc) and GSS algorithms are given. To create and simulate the suggested system, MATLAB and its Simulink program were utilized.

The simulation results demonstrate that the proposed system, using MATLAB/Simulink, effectively evaluates the robustness of the PV array characteristics and the PVAE control approach. Under varying solar irradiations, the GSS and INC MPPT techniques exhibit superior performance in tracking, stability, and response times compared to the P&O algorithm. The GSS technique, in particular, ensures quick system responses, enhancing PVPS efficiency. Experimental validation confirms the emulator's ability to accurately replicate solar panel behavior, providing a viable solution for sustainable water pumping in agriculture and irrigation.

Future work could focus on integrating advanced machine learning algorithms to further optimize MPPT performance under diverse and unpredictable weather conditions. Additionally, expanding the system to include hybrid energy sources, such as wind or battery storage, could enhance reliability and efficiency. Field testing the emulator in real-world agricultural settings would provide valuable insights and help refine the control strategies for broader applications.

## Author Contributions

**Conceptualization:** Amir Fatah, Tarek Boutabba, Idriss Benlaloui, Mohamed Metwally Mahmoud, Wulfran Fendzi Mbasso.

**Funding acquisition:** Wulfran Fendzi Mbasso.

**Investigation:** Idriss Benlaloui, Mohamed Metwally Mahmoud, Wulfran Fendzi Mbasso, Hany S. Hussein.

**Methodology:** Amir Fatah, Tarek Boutabba, Idriss Benlaloui, Wulfran Fendzi Mbasso.

**Resources:** Mohamed Metwally Mahmoud.

**Software:** Amir Fatah, Tarek Boutabba, Wulfran Fendzi Mbasso.

**Supervision:** Said Drid.

**Validation:** Mohamed Metwally Mahmoud, Mahmoud M. Hussein, Wulfran Fendzi Mbasso, Ahmed M. Ewias.

**Visualization:** Mahmoud M. Hussein, Ahmed M. Ewias.

**Writing – original draft:** Amir Fatah, Wulfran Fendzi Mbasso.

**Writing – review & editing:** Tarek Boutabba, Idriss Benlaloui, Said Drid, Mohamed Metwally Mahmoud, Mahmoud M. Hussein, Hany S. Hussein, Ahmed M. Ewias.

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
