## [Decision Letter · Decision Letter 0]

8 May 2024

PONE-D-24-13352Design, and Dynamic Evaluation of a Novel Photovoltaic Pumping System Emulation with DS1104 Hardware Setup: Towards Innovative in Green Energy SystemsPLOS ONE

Dear Dr. FENDZI MBASSO,

Thank you for submitting your manuscript to PLOS ONE. After careful consideration, we feel that it has merit but does not fully meet PLOS ONE’s publication criteria as it currently stands. Therefore, we invite you to submit a revised version of the manuscript that addresses the points raised during the review process.

We look forward to receiving your revised manuscript.

Kind regards,

Hossein Abedini, Ph.D.

Academic Editor

PLOS ONE

Journal Requirements:

4. Please ensure that you refer to Figure 7 and 8 in your text as, if accepted, production will need this reference to link the reader to the figure.

5. We note you have included a table to which you do not refer in the text of your manuscript. Please ensure that you refer to Table 2 in your text; if accepted, production will need this reference to link the reader to the Table.

Reviewers' comments:

Reviewer's Responses to Questions

**Comments to the Author**

1. Is the manuscript technically sound, and do the data support the conclusions?

Reviewer #1: Yes

Reviewer #2: Yes

Reviewer #3: Yes

Reviewer #4: Partly

Reviewer #5: Yes

2. Has the statistical analysis been performed appropriately and rigorously? 

Reviewer #1: Yes

Reviewer #2: N/A

Reviewer #3: No

Reviewer #4: I Don't Know

Reviewer #5: Yes

3. Have the authors made all data underlying the findings in their manuscript fully available?

Reviewer #1: Yes

Reviewer #2: No

Reviewer #3: Yes

Reviewer #4: Yes

Reviewer #5: Yes

4. Is the manuscript presented in an intelligible fashion and written in standard English?

Reviewer #1: Yes

Reviewer #2: Yes

Reviewer #3: Yes

Reviewer #4: No

Reviewer #5: Yes

5. Review Comments to the Author

Reviewer #1: The manuscript proposes "Design, and Dynamic Evaluation of a Novel Photovoltaic Pumping System Emulation

with DS1104 Hardware Setup: Towards Innovative in Green Energy Systems"

The reviewer's concerns are as follows:

1- The novelty of this study is not clear. The introduction section should include a flowchart summarizing the article.

2- Authors have not provided sufficient comparison between their proposed methodology and the previous ones in the introduction. According to the current introduction, there is no contribution in the current study.

3- "It is recommended to add the following references.

a)Adak, S. Harmonics Mitigation of Stand-Alone Photovoltaic System Using LC Passive Filter. J. Electr. Eng. Technol. 16, 2389–2396 (2021). https://doi.org/10.1007/s42835-021-00777-7

b)Adak, S., Cangi, H. Development software program for finding photovoltaic cell open-circuit voltage and fill factor based on the photovoltaic cell one-diode equivalent circuit model. Electr Eng 106, 1251–1264 (2024). https://doi.org/10.1007/s00202-023-02082-0

Reviewer #2: This is a very good research effort by the authors especially from the experimental setup point of view. However, the following observations will improve the quality of this work

(a) Author may need to work on an extensive literature review section in this work to affirm the background and indicate the extent of work done by similar authors to reflect the novelty of this work

(b) Green energy systems concepts is missing completely, its relationship to the experiment and comparison of various green energy system with emphasis on the PV pumping system will improve the overall quality of this work with respect to modern innovations in energy systems.

(c) An improved quality of the image presented in figure 9 will go a long way

(d) The Laboratory setup image in figure 10 can also be more clearer or presented first in segment before the overall setup presentation.

(e) Authors have done a very good job proposing a frame work following thier consideration of PV array characteristics and PVAE Control approach albeit the simulation results are missing in the abstracts and the conclusion sections of the work.

(f) The graphs presented in the experimental and simulation results are not well discussed in line with the objectives especially when it comes to optimization control strategy.

Overall it is a good work and well put together.

Reviewer #3: This study introduces an emulation approach for photovoltaic (PV) Water Pumping(WP) systems.

In this work integrates two emulators into a single chain, effectively combining PV and WP

emulation while optimizing system dynamics. The goal to devise a comprehensive emulator for PV

water pumping systems and assess an improved control algorithm. To regulate the water MP, we

deploy an optimized scalar control strategy. Validation of this control strategy is conducted through

an experimental arrangement utilizing the dSPACE control desk DS1104. Obtained results affirm

the emulator's proficiency in faithfully reproducing genuine solar panel characteristics.

The following are comments/suggestions to improve the manuscript:

1. It seems the authors have some positive results. Please compare the newly adopted power point

tracking algorithm( GSS) with the conventional one (perturb and observe) ; show the improvement

on the power plots (on the same plots).

2. Nomenclature is should be written clearly. Some lines are miss-aligned or spelled wrongly.

3. Your figures are not in good quality/resolution(figure 4,7, and 9). Please use improved quality images

for your simulation and experiments.

4. The literature review is not sufficient. Include the following papers on converter control, inverter

and motor control as such as

- Improved Model Predictive Speed Control of a PMSM via Laguerre Functions, Mathematical

Problems in Engineering, vol. 2024, Article ID 5562771, 14 pages, 2024. https://doi.org/

10.1155/2024/5562771.

- Stabilization and Voltage Regulation of the Buck DC-DC Converter Using Model Predictive of

Laguerre Functions, Studies in Informatics and Control, ISSN 1220-1766, vol. 26(3), pp. 315-324,

2017. https://doi.org/10.24846/v26i3y201707

- A Modified Controller for Three Level Three-Phase Voltage Source Inverter based on Laguerre

Functions. International Journal of Computer Applications, 182(25), 21–28. https://doi.org/

10.5120/ijca2018918081.

5. There are typos Typos.

- In the abstract, The first one is a PV system emulator that employs back buck converter control to

faithfully mirror the characteristics of PV panels.

6. Please write in passive speech. Avoid pronouns we, their, our, them etc.

7. Lastly not least, plot the simulation and experimental results on the same figure/plots so that your

experiments can validate the simulation results.

Reviewer #4: The word emulation is not the proper one in the title and in the text because your research concerns the PV pumping system emulator, not emulation. Please correct this. The abstract should refer only to the main idea of the research presented in the manuscript not to other introduction - the context of your research is already known. Also a lot of well-known information in your text. The focus on the main problem - developing a new emulator should be in the center, not else. A comparison should be more interesting and could reveal the novelty of your research. In my opinion chapter 4 is welcome and the obtained results by simulation and experiment must be compared with existing emulators.The novelty must be better exaplained. The Conclusion chapter is also important.. The manuscript should be improved.

Reviewer #5: The manuscript has a lot of simulation and experiment findings. We do need to know more about how to understand these results, though. For instance, the conversation could go into more detail about how different control methods (like P&O, INC, and GSS) affect the performance of the emulation system when the amount of sunlight changes.

Some technical details about how the emulator is set up and how the control methods work could be made clearer. The technical depth of the text would be improved by explaining how the boost converter with MPPT algorithm works and how it affects the efficiency of power tracking.

The manuscript might discuss more about what the suggested emulation system means in the real world, especially when it comes to agriculture and irrigation.

6. PLOS authors have the option to publish the peer review history of their article (what does this mean?). If published, this will include your full peer review and any attached files.

Reviewer #1: No

Reviewer #2: No

Reviewer #3: No

Reviewer #4: No

Reviewer #5: **Yes: **Noramalina Abdullah

---

## [Author Response · Author response to Decision Letter 0]

28 Jun 2024

To,

The Editor-in-Chief and Reviewers PLOS ONE

Subject: Submission of revised manuscript on “Design, and Dynamic Evaluation of a Novel Photovoltaic Pumping System Emulation with DS1104 Hardware Setup: Towards Innovative in Green Energy Systems”

Ref: # PONE-D-24-11483

 Dear Editor and Reviewers,

The authors would like to thank the editor and the respected reviewers for their precious time and invaluable and constructive comments on the work entitled “Design, and Dynamic Evaluation of a Novel Photovoltaic Pumping System Emulation with DS1104 Hardware Setup: Towards Innovative in Green Energy Systems”. Indeed, we are pleased that our paper has found a positive echo in your journal. We have carefully taken into account all constructive comments and recommendation addressed by all the respected reviewers. The corresponding changes and refinements have been carried out in the revised paper according to all the comments and recommendations of the respected reviewers separately one by one.

In the revised version of the paper, all changes and additions are clearly marked. Additionally, minor corrections within the body text are highlighted in red. If a figure or table has been edited or added, the caption is also highlighted in red. We have addressed all comments from the respected reviewers in a point-by-point manner, providing separate responses for each reviewer.

We hope that the revised manuscript has been improved to reach the required satisfaction for publication in your esteemed journal. We are eagerly waiting for your kind feedback and decision.

Please feel free to contact us if there are any further requirements to be carried out.

Thank you again for your kind patience and attention.

Sincerely Yours,

Wulfran FENDZI MBASSO,

Email of the Corresponding Author: fendzi.wulfran@yahoo.fr

 Comment and response

The responses and corrections concerning the comments and remarks have been addressed carefully in the revised version of the paper.

The manuscript proposes "Design, and Dynamic Evaluation of a Novel Photovoltaic Pumping System Emulation with DS1104 Hardware Setup: Towards Innovative in Green Energy Systems".

The reviewer's concerns are as follows:

Comment 1: The novelty of this study is not clear. The introduction section should include a flowchart summarizing the article.

Authors’ response to comment 1: Thank you for this important comment. The introduction has been revised in accordance with the suggested points. We kindly invite the respected reviewer to verify these updates within the revised version of the paper.

Comment 2: Authors have not provided sufficient comparison between their proposed methodology and the previous ones in the introduction. According to the current introduction, there is no contribution in the current study.

Authors’ response to comment 2: We would like to thank the respected reviewer for this point. Accordingly, the introduction has been revised as per the suggested points. We have included subsection 4.2.2 titled "Comparative Studies" to showcase the performance of the techniques employed. We kindly request the esteemed reviewer to review these updates in the revised version of the paper.

We sincerely thank the reviewer for their invaluable comments, which have significantly enhanced the quality of our manuscript.

Comment 3: It is recommended to add the following references:

a) Adak, S. Harmonics Mitigation of Stand-Alone Photovoltaic System Using LC Passive Filter. J. Electr. Eng. Technol. 16, 2389–2396 (2021). https://doi.org/10.1007/s42835-021-00777-7.

b) Adak, S., Cangi, H. Development software program for finding photovoltaic cell open-circuit voltage and fill factor based on the photovoltaic cell one-diode equivalent circuit model. Electr Eng 106, 1251–1264 (2024). https://doi.org/10.1007/s00202-023-02082-0.

Authors’ response to comment 3: Thank you for your valuable suggestions. We have incorporated the two references into our revised manuscript. These additions will enhance the completeness and credibility of our study.

Response to Reviewer 2

The responses and corrections concerning the comments and remarks of the Reviewer 2 have been addressed carefully in the revised version of the paper.

This is a very good research effort by the authors especially from the experimental setup point of view. However, the following observations will improve the quality of this work.

Authors’ appreciation on over all comments of the reviewer of this manuscript. The authors would like to thank the respected Reviewer for this very encouraging appreciation

Comment 1: Author may need to work on an extensive literature review section in this work to affirm the background and indicate the extent of work done by similar authors to reflect the novelty of this work.

Authors’ response to comment 1: Thank you very much for your detailed and constructive feedback on our manuscript. We greatly appreciate your insights and have made significant efforts to address your concerns in the revised version. 

We acknowledge that the original manuscript may have lacked optimal organization. We have revised the structure to improve the clarity and coherence of our arguments. Concerning the literature review, we have expanded our literature review to clearly highlight the differences between our work and the current literature. By explicitly identifying gaps in existing research, we aim to clarify our contribution to the field. For the reorganization of the literature review, we have reorganized this section to ensure a more fluent presentation of the information. 

[1] Adak, S. Harmonics Mitigation of Stand-Alone Photovoltaic System Using LC Passive Filter. J. Electr. Eng. Technol. 16, 2389–2396 (2021). https://doi.org/10.1007/s42835-021-00777-7.

[2] Adak, S., Cangi, H. Development software program for finding photovoltaic cell open-circuit voltage and fill factor based on the photovoltaic cell one-diode equivalent circuit model. Electr Eng 106, 1251–1264 (2024). https://doi.org/10.1007/s00202-023-02082-0.

[3] Improved Model Predictive Speed Control of a PMSM via Laguerre Functions, Mathematical Problems in Engineering, vol. 2024, Article ID 5562771, 14 pages, 2024. https://doi.org/ 10.1155/2024/5562771.

[4] Stabilization and Voltage Regulation of the Buck DC-DC Converter Using Model Predictive of Laguerre Functions, Studies in Informatics and Control, ISSN 1220-1766, vol. 26(3), pp. 315-324, 2017. https://doi.org/10.24846/v26i3y201707.

[5] A Modified Controller for Three Level Three-Phase Voltage Source Inverter based on Laguerre Functions. International Journal of Computer Applications, 182(25), 21–28. https://doi.org/10.5120/ijca2018918081.

Comment 2: Green energy systems concepts is missing completely, its relationship to the experiment and comparison of various green energy system with emphasis on the PV pumping system will improve the overall quality of this work with respect to modern innovations in energy systems.

Authors’ response to comment 2: Thank you for this comment. It has been covered in the introduction part of the updated part. kindly invite the respected reviewer to verify these updates within the revised version of the paper. 

Comment 3: An improved quality of the image presented in figure 9 will go a long way.

Authors’ response to comment 3: Thank you for this comment. We have improved the quality of the image presented in Figure to ensure better visibility and comprehension for the readers. We kindly invite the respected reviewer to verify these updates within the revised version of the paper. 

Comment 4: The Laboratory setup image in figure 10 can also be clearer or presented first in segment before the overall setup presentation.

Authors’ response to comment 4: Thank you for this comment. We have improved the quality of the laboratory setup image in figure 10 to ensure better visibility and comprehension for the readers. We kindly invite the respected reviewer to verify these updates within the revised version of the paper. 

Comment 5: Authors have done a very good job proposing a frame work following their consideration of PV array characteristics and PVAE Control approach albeit the simulation results are missing in the abstracts and the conclusion sections of the work.

Authors’ response to comment 5: Once again, thank you for your positive feedback on our proposed framework considering the PV array characteristics and PVAE control approach. We appreciate your observation regarding the absence of simulation results in the abstract and conclusion sections. Accordingly, we have revised these sections to include a summary of the key simulation results, ensuring that the impact and effectiveness of our proposed techniques are clearly communicated. We kindly invite the respected reviewer to verify these updates within the revised version of the paper.

Comment 6: The graphs presented in the experimental and simulation results are not well discussed in line with the objectives especially when it comes to optimization control strategy.

Authors’ response to comment 6: Thank you for your valuable feedback. The objective of this work is to integrate both the PV and water pump emulators into a cohesive system, validated through experimental setups using the dSPACE control desk DS1104, which accurately reproduces solar panel characteristics. Furthermore, the integration of a new MPPT algorithm based on the GSO technique aims to optimize the MPPT control strategy. In response to your comments, we have included a new subsection 4.2.2 titled "Comparative Studies" to showcase the performance of the techniques employed. We kindly request the esteemed reviewer to review these updates in the revised version of the paper.

Overall it is a good work and well put together.

Once again, thank you for your invaluable comments. They have been instrumental in improving the quality of our manuscript. 

Response to Reviewer 3

The responses and corrections concerning the comments and remarks of the Reviewer 3 have been addressed carefully in the revised version of the paper.

This study introduces an emulation approach for photovoltaic (PV) Water Pumping (WP) systems.

In this work integrates two emulators into a single chain, effectively combining PV and WP

emulation while optimizing system dynamics. The goal to devise a comprehensive emulator for PV water pumping systems and assess an improved control algorithm. To regulate the water MP, we deploy an optimized scalar control strategy. Validation of this control strategy is conducted through an experimental arrangement utilizing the dSPACE control desk DS1104. Obtained results affirm the emulator's proficiency in faithfully reproducing genuine solar panel characteristics.

The following are comments/suggestions to improve the manuscript.

Thank you for your thorough review and constructive comments on our manuscript. We are committed to addressing all the main drawbacks outlined in your comments and have considered your suggestions to enhance the overall quality of the manuscript.

Comment 1: It seems the authors have some positive results. Please compare the newly adopted power point tracking algorithm (GSS) with the conventional one (perturb and observe); show the improvement on the power plots (on the same plots). 

Authors’ response to comment 1: Thank you for this constructive comment. Accordingly, we have added subsection 4.2.2 titled "Comparative Studies" to showcase the performance of the techniques employed. We kindly request the esteemed reviewer to review these updates in the revised version of the paper.

Comment 2: Nomenclature is should be written clearly. Some lines are miss-aligned or spelled wrongly.

Authors’ response to comment 2: Thank you for drawing our attention to this point. Consequently, we have proofread the manuscript to rectify any typos and clarify any abbreviations. The whole paper has been thoroughly reviewed for this matter. We kindly invite the respected reviewer to verify these updates within the revised version of the paper.

Comment 3: Your figures are not in good quality/resolution (figure 4,7, and 9). Please use improved quality images for your simulation and experiments.

Authors’ response to comment 3: Thank you for this comment. We have improved the quality of images to ensure better visibility and comprehension for the readers. We kindly invite the respected reviewer to verify these updates within the revised version of the paper. 

Comment 4: The literature review is not sufficient. Include the following papers on converter control, inverter and motor control as such as:

[1] Improved Model Predictive Speed Control of a PMSM via Laguerre Functions, Mathematical Problems in Engineering, vol. 2024, Article ID 5562771, 14 pages, 2024. https://doi.org/ 10.1155/2024/5562771.

[2] Stabilization and Voltage Regulation of the Buck DC-DC Converter Using Model Predictive of Laguerre Functions, Studies in Informatics and Control, ISSN 1220-1766, vol. 26(3), pp. 315-324, 2017. https://doi.org/10.24846/v26i3y201707.

[3] A Modified Controller for Three Level Three-Phase Voltage Source Inverter based on Laguerre Functions. International Journal of Computer Applications, 182(25), 21–28. https://doi.org/10.5120/ijca2018918081.

Authors’ response to comment 4: Thank you for your feedback on our manuscript. We appreciate your suggestion and have incorporated the recommended references into our revised paper to enhance the literature review. We kindly invite the esteemed reviewer to verify these updates in the revised version of the paper.

Comment 5: There are typos.

- In the abstract, the first one is a PV system emulator that employs back buck converter control to faithfully mirror the characteristics of PV panels.

Authors’ response to comment 5: Thank you for drawing our attention to this point. Consequently, we have proofread the manuscript to rectify any typos. The whole paper has been thoroughly reviewed for this matter. We kindly invite the respected reviewer to verify these updates within the revised version of the paper.

Comment 6: Please write in passive speech. Avoid pronouns we, their, our, them etc.

Authors’ response to comment 6: Thank you for bringing this to our attention. As a result, the manuscript has been proofread to eliminate pronouns such as we, their, our, and them by rewriting in passive voice. The entire paper has been thoroughly reviewed in this regard. We kindly invite the esteemed reviewer to verify these updates in the revised version of the paper.

Comment 7: Lastly not least, plot the simulation and experimental results on the same figure/plots so that your experiments can validate the simulation results.

Authors’ response to comment 7: We appreciate the suggestion. However, due to the inherent differences in experimental conditions, direct comparison on the same plots isn't feasible. We have already presented simulation and experimental results separately to accurately reflect their respective contexts and findings.

Response to Reviewer 4

The responses and corrections concerning the comments and remarks of the Reviewer 4 have been addressed carefully in the revised version of the paper.

Comment 1: The manuscript has a lot of simulation and experiment findings. We do need to know more about how to understand these results, though. For instance, the conversation could go into more detail about how different control methods (like P&O, INC, and GSS) affect the performance of the emulation system when the amount of sunlight changes.

Authors’ response to comment 1: We would like to thank the respected reviewer for this point. Accordingly, we have included subsection 4.2.2 titled "Comparative Studies" to showcase the performance of the techniques employed. We kindly request the esteemed reviewer to review these updates in the revised version of the paper.

Comment 2: Some technical details about how the emulator is set up and how the control methods work could be made clearer. The technical depth of the text would be improved by explaining how the boost converter with MPPT algorithm works and how it affects the efficiency of power tracking.

Auth

---

## [Decision Letter · Decision Letter 1]

19 Jul 2024

Design, and Dynamic Evaluation of a Novel Photovoltaic Pumping System Emulation with DS1104 Hardware Setup: Towards Innovative in Green Energy Systems

PONE-D-24-13352R1

Dear Dr. FENDZI MBASSO,

We’re pleased to inform you that your manuscript has been judged scientifically suitable for publication and will be formally accepted for publication once it meets all outstanding technical requirements.

Kind regards,

Hossein Abedini, Ph.D.

Academic Editor

PLOS ONE

Additional Editor Comments (optional):

Comments from PLOS Editorial Office: We note that one or more reviewers has recommended that you cite specific previously published works in an earlier round of revision. As always, we recommend that you please review and evaluate the requested works to determine whether they are relevant and should be cited. It is not a requirement to cite these works and you may remove them before the manuscript proceeds to publication. We appreciate your attention to this request.

Reviewers' comments:

Reviewer's Responses to Questions

**Comments to the Author**

1. If the authors have adequately addressed your comments raised in a previous round of review and you feel that this manuscript is now acceptable for publication, you may indicate that here to bypass the “Comments to the Author” section, enter your conflict of interest statement in the “Confidential to Editor” section, and submit your "Accept" recommendation.

Reviewer #1: All comments have been addressed

Reviewer #3: All comments have been addressed

2. Is the manuscript technically sound, and do the data support the conclusions?

Reviewer #1: Yes

Reviewer #3: Yes

3. Has the statistical analysis been performed appropriately and rigorously? 

Reviewer #1: Yes

Reviewer #3: Yes

4. Have the authors made all data underlying the findings in their manuscript fully available?

Reviewer #1: Yes

Reviewer #3: Yes

5. Is the manuscript presented in an intelligible fashion and written in standard English?

Reviewer #1: Yes

Reviewer #3: Yes

6. Review Comments to the Author

Reviewer #1: (No Response)

Reviewer #3: The authors addressed all the comments and suggestions.

The reviewer don't have any more concerns.

7. PLOS authors have the option to publish the peer review history of their article (what does this mean?). If published, this will include your full peer review and any attached files.

Reviewer #1: No

Reviewer #3: No

---

## [Editor Report · Acceptance letter]

14 Aug 2024

PONE-D-24-13352R1 

PLOS ONE

Dear Dr. Fendzi Mbasso, 

I'm pleased to inform you that your manuscript has been deemed suitable for publication in PLOS ONE. Congratulations! Your manuscript is now being handed over to our production team.

Kind regards, 

on behalf of

Dr. -Ing Hossein Abedini 

Academic Editor

PLOS ONE